# Adaptive DNA Sequence Modeling via Synergistic Plasticity Units

**Binghao Liu** [1 2]   **Wenzheng Zhao** [1 2]   **Zhijie Zheng** [1 2]   **Fei Gu** [1 2]

## Abstract

Effective DNA modeling demands the integration of complex patterns such as local motifs, long-range dependencies, and periodic signals. Yet, architectures like CNNs, Transformers, and SSMs are hindered by static or time-domain-exclusive designs, which limit their representational flexibility. To address this, we introduce the **Synergistic Plasticity Unit (SPU)**, a scalable architecture that achieves multi-level plasticity through three synergistic layers. Specifically, SPU integrates a *Locus Plasticity Layer* (LPL) to capture fine-grained local motifs via token-specific convolution operations, while utilizing a *Domain Plasticity Layer* (DPL) to form multi-domain global features by concurrently modeling sequential (time) and spectral (frequency) patterns. Furthermore, it incorporates a *Saliency Plasticity Layer* (SPL) to optimize information flow through dual-axis saliency scoring. Supported by theoretical analysis, extensive empirical validation, and in-depth biological interpretation, this unified design enables SPU to achieve state-of-the-art performance with quasi-linear complexity, establishing a robust and principled paradigm for DNA modeling.

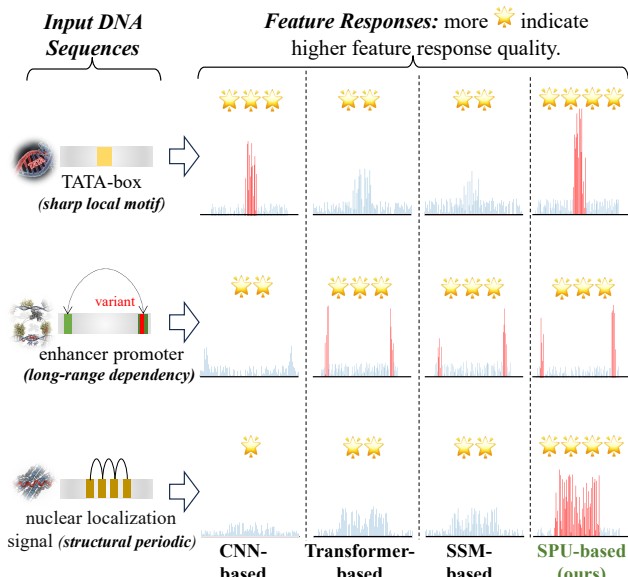

*Figure 1.* **Comparison of feature responses to diverse DNA patterns across different models.** CNN-based models are adept at local feature extraction but are structurally constrained in capturing long-range and periodic patterns. Transformer-based and SSM-based models excel at capturing global dependencies but show limited efficacy in modeling local motifs and frequency-sensitive signals. In contrast, the SPU-based model utilizes a multi-level plasticity mechanism to achieve a holistic representation that integrates local, global, and frequency-sensitive features.

## 1. Introduction

Genomic sequence analysis presents a significant challenge due to the extraordinary length and informational complexity of DNA. The advancement of computational models, however, offers a powerful solution. Leveraging their efficiency and precision, these models are uniquely suited to process vast and intricate biological data. Consequently, applying computational methods to genomic analysis has become an increasingly prevalent and fruitful approach, showing great promise in deciphering the complexities of the genome (Bellot et al., 2018; Amin et al., 2018; Eraslan et al., 2019; Zou et al., 2019; Tian et al., 2019; Wang et al., 2020b; Talukder et al., 2021).

DNA sequences exhibit a profound informational diversity, characterized by sparse yet critical features, such as local motifs (Wasserman & Sandelin, 2004), long-range dependencies (Schoenfelder & Fraser, 2019), and periodic patterns (Brogaard et al., 2012). However, existing DNA foundation models are constructed upon common architectures, such as Convolutional Neural Networks (CNNs) (Zhou & Troyanskaya, 2015; Bo et al., 2025), Transformers (Zhou et al., 2024a; Dalla-Torre et al., 2025), and State Space Models (SSMs) (Schiff et al., 2024). These architectures rely on basic modules with static or single-domain operations, which limit their architectural flexibility in capturing diverse features inherent in DNA sequences. CNNs, while effective at identifying local motifs, struggle to capture long-

[1]DAMO Academy, Alibaba Group, Hangzhou, 310023, China [2]Hupan Lab, Hangzhou, 310023, China. Correspondence to: Fei Gu <gufei.gf@alibaba-inc.com>.

*Proceedings of the 43rd International Conference on Machine Learning*, Seoul, South Korea. PMLR 306, 2026. Copyright 2026 by the author(s).

range dependencies due to their static kernels and limited receptive fields (Kelley et al., 2016; Liu et al., 2022; Yu & Wang, 2025). Transformers (Vaswani et al., 2017) excel at modeling long-range interactions through the attention mechanism, but lack the strong inductive biases for local patterns inherent to CNNs, making it more challenging to effectively model local features for DNA sequences (Avsec et al., 2021). Moreover, their quadratic complexity is computationally prohibitive for genome-scale sequences. Although SSMs (e.g., Mamba) achieve linear complexity, their lossy state compression may lose fine-grained details (Gu & Dao, 2023). Other works (Avsec et al., 2021; Yang et al., 2025) adopt hybrid architectures as a pragmatic compromise, but still fall short of fundamentally addressing DNA's inherent multi-scale complexity.

A deeper limitation is that these models operate exclusively in the time domain and thus lack the intrinsic mechanisms to directly model frequency-sensitive features. Consequently, vital periodic signals (including codon periodicity (Yin & Yau, 2007), nucleosome positioning signals (Brogaard et al., 2012), and tandem repeats (Benson, 1999)) remain poorly represented by current architectures, as illustrated in Fig. 1.

To address these limitations, we propose the **Synergistic Plasticity Unit (SPU)**, a novel scalable building unit designed by the principle of multi-level plasticity. The unit begins with the Locus Plasticity Layer (LPL), which dynamically extracts robust local features from DNA embeddings. It achieves this by using varying-sized convolutional kernel bases to capture multi-scale meta features (fundamental features used for token-level feature construction), and generating token-specific weights through a conv layer to combine those meta features into fine-detailed representations. These representations are then processed in parallel by the Domain Plasticity Layer (DPL) and the Saliency Plasticity Layer (SPL). The DPL employs a global convolution and a wavelet transform to fuse sequential patterns with spectral features, allowing it to model both global dependencies and frequency-sensitive representations. Simultaneously, the SPL computes saliency scores across both the channel and position axes, identifying the most informative elements. Finally, the DPL's time-frequency representations and the SPL's saliency scores are combined via element-wise multiplication, which selectively amplifies critical information while attenuating noise. Through the synergistic interplay of these three layers, the SPU organically integrates local-global, time-frequency domain information, achieving a holistic capture of features and dependencies within DNA sequences, as shown in Fig. 1. Furthermore, SPU can be stacked to construct various-sized DNA foundation models, termed **SPU-DNA**.

Across a wide array of benchmarks testing both short- and long-range sequence dependencies, our method establishes a new state-of-the-art, outperforming previous methods by a notable margin. Extensive theoretical derivations and in-depth model analyses collectively validate the advantages of our approach, demonstrating its effectiveness in terms of theoretical soundness, experimental performance, and biological interpretability.

The contributions of this work are listed below:

- We introduce the Synergistic Plasticity Unit (SPU), a novel and scalable building unit for DNA sequence analysis. Its design, centered on multi-level plasticity, allows it to adaptively adjust its computational strategy to capture the diverse patterns inherent in DNA.

- We design the Locus Plasticity Layer (LPL) to capture fine-grained local patterns through a token-specific extraction mechanism; the Domain Plasticity Layer (DPL) to jointly model and fuse time-frequency representations using a global convolution and a wavelet transform; and the Saliency Plasticity Layer (SPL) to learn dynamic saliency scores along channel and position axes for feature refining.

- The SPU-based DNA model (SPU-DNA) achieves state-of-the-art performance across various benchmarks while remaining highly efficient in terms of both parameter count and computational complexity, delivering multifaceted contributions in theoretical rigor, empirical excellence, and biological interpretability that collectively establish it as a leading paradigm in DNA sequence modeling.

## 2. Related Work

**DNA Sequence Analysis** Analyzing the vast information encoded within DNA sequences is a fundamental challenge in modern genomics. This genetic information, spanning both coding and non-coding regions, dictates everything from protein synthesis to the complex orchestration of gene expression, making it critical for understanding health and disease. Traditional bioinformatic methods, which are largely statistical in nature, often fail to capture the intricate, context-dependent patterns in DNA sequences. Deep learning methods, such as CNNs (He et al., 2016), Transformers (Vaswani et al., 2017), and SSMs (Gu & Dao, 2023), have driven a paradigm shift by learning hierarchical features directly from raw data, overcoming these limitations.

**Time and Frequency Modeling** Signal analysis bifurcates into time-domain methods (e.g., (Schafer, 2011)) that capture local dynamics, and frequency-domain techniques like FFT (Nussbaumer, 1981) that identify global periodicity but lack localization. Wavelets (Bentley & McDonnell, 1994) offer multi-resolution time-frequency analysis for non-stationary signals. Wavelet-CNN (Fujieda et al.,

2018) processes signals using wavelet bases with relatively fixed scales. This parallel extends to deep learning: CNNs act as learnable time-domain filters, whereas models like Hyena (Nguyen et al., 2023) leverage global convolutions accelerated by FFT.

**CNN-based DNA models** As a foundational architecture in genomics, CNNs remain a cornerstone of the field. Early models like DeepBind (Alipanahi et al., 2015) applied CNNs to predict protein-binding specificities, while DeepSEA (Zhou & Troyanskaya, 2015) predicted the functional effects of non-coding variants. CondConv (Yang et al., 2019) introduced dynamic convolutions to enhance flexibility, but its dynamic adaptation is limited to the sample level. CBAM (Woo et al., 2018) uses spatial and channel attention to filter information for images, but it is not adapted for DNA. More recently, architectures have adapted CNNs to capture long-range dependencies. Hyena (Nguyen et al., 2023), for example, employed long convolutions to model million-token sequences, while ConvNova (Bo et al., 2025) incorporated advanced features like dilated and gated convolutions to achieve a wider receptive field.

**Transformer-based DNA models** The introduction of the Transformer architecture marked a pivotal moment in genomics, reframing sequence analysis by treating DNA as a language. DNABERT (Ji et al., 2021) pioneered this approach by introducing a k-mer tokenization strategy. Subsequent iterations improved upon this foundation: DNABERT-2 (Zhou et al., 2024a) enhanced computational efficiency with Byte Pair Encoding (BPE), while DNABERT-S (Zhou et al., 2024b) employed specialized contrastive learning for species-aware embeddings. This foundation model concept was significantly scaled up with the Nucleotide Transformer (NT) (Dalla-Torre et al., 2025), featuring models with up to 2.5 billion parameters. Hybrid architectures have also emerged, such as Enformer (Avsec et al., 2021), which integrated convolutional layers with an attention mechanism to predict gene expression from DNA sequences. Concurrently, models like Space (Yang et al., 2025) utilized large Mixture-of-Experts (MoE) (Yuksel et al., 2012; Masoudnia & Ebrahimpour, 2014; Lin et al., 2024) to learn DNA representations via supervised training on genomic profiles.

**SSM-based DNA models** State Space Models (Hamilton, 1994; Paninski et al., 2010), particularly the Mamba architecture, have recently emerged as a powerful and efficient alternative for DNA modeling. Caduceus (Schiff et al., 2024) introduced the MambaDNA block, which enforces reverse-complement equivariance, a critical inductive bias for genomics. Among early state transition models, DanQ (Quang & Xie, 2016) was an effective model that combined a bidirectional LSTM (BiLSTM) network with a CNN for DNA analysis. More recently, Bio-xLSTM (Schmidinger et al., 2025) adapted the xLSTM architecture for DNA and protein

sequence modeling, achieving high efficiency and accuracy. Prism (Yang et al., 2026) warns that long-context DNA can introduce confounding noise, highlighting the need for more effective information filtering.

Despite significant progress, key challenges remain unresolved across these architectures. While CNN-based models are efficient, their ability to model long-range dependencies remains limited. Transformer-based models excel at modeling global dependencies, but they are less effective at directly capturing local features and are hampered by a quadratic computational complexity that is prohibitive for ultra-long DNA sequences. SSM-based models achieve linear complexity but risk losing fine-grained details by compressing information into a finite state. More critically, all these methods operate almost exclusively in the time domain. This shared limitation means they lack a dedicated mechanism to directly model frequency-sensitive patterns, such as nucleosome periodicity, which are crucial in genomics and cannot be effectively leveraged by existing models.

## 3. Methodology

In this section, we first present the core design of the SPU, substantiated by a theoretical analysis. We then describe how to utilize SPU to construct DNA foundation models of varying scales. Finally, we outline the pre-training and fine-tuning details of the SPU-based DNA models.

### 3.1. Synergistic Plasticity Unit

As shown in Fig. 2(left), the SPU consists of the Locus Plasticity Layer (LPL), the Domain Plasticity Layer (DPL), and the Saliency Plasticity Layer (SPL). The LPL first processes an input DNA embedding ($\mathbf{E}$) with token-specific feature extraction, yielding a contextually-rich output $\mathbf{F}^{\text{lpl}}$. This output is then fed in parallel to the DPL and SPL. The DPL produces unified time-frequency features ($\mathbf{F}^{\text{dpl}}$), while the SPL computes saliency scores across the channel and position axes. The saliency scores and the time-frequency features are combined to obtain the SPU's output.

#### 3.1.1. LOCUS PLASTICITY LAYER

As illustrated in Fig. 2(right), the LPL is comprised of two parallel branches: a bank of convolutional kernel bases and a coefficient generation branch that employs a $1 \times 1$ convolution. The kernel bases contain a set of $T$ distinct convolutional kernels, $\mathcal{K} = \{k_0, k_1, \ldots, k_{T-1}\}$ (each with a different size), which are referred to as static "bases". Given an input DNA embedding $\mathbf{E} \in \mathbb{R}^{B \times L \times D}$, where $B$ is the batch size, $L$ is the sequence length and $D$ is the embedding dimension, the kernel bases branch applies each kernel $k_i$ to $\mathbf{E}$ to produce a group of meta features, $\mathbf{F}_g = \{\mathbf{F}_0, \mathbf{F}_1, \ldots, \mathbf{F}_{T-1}\}$, where $\mathbf{F}_i = \text{Conv}(k_i, \mathbf{E}), \quad i =$

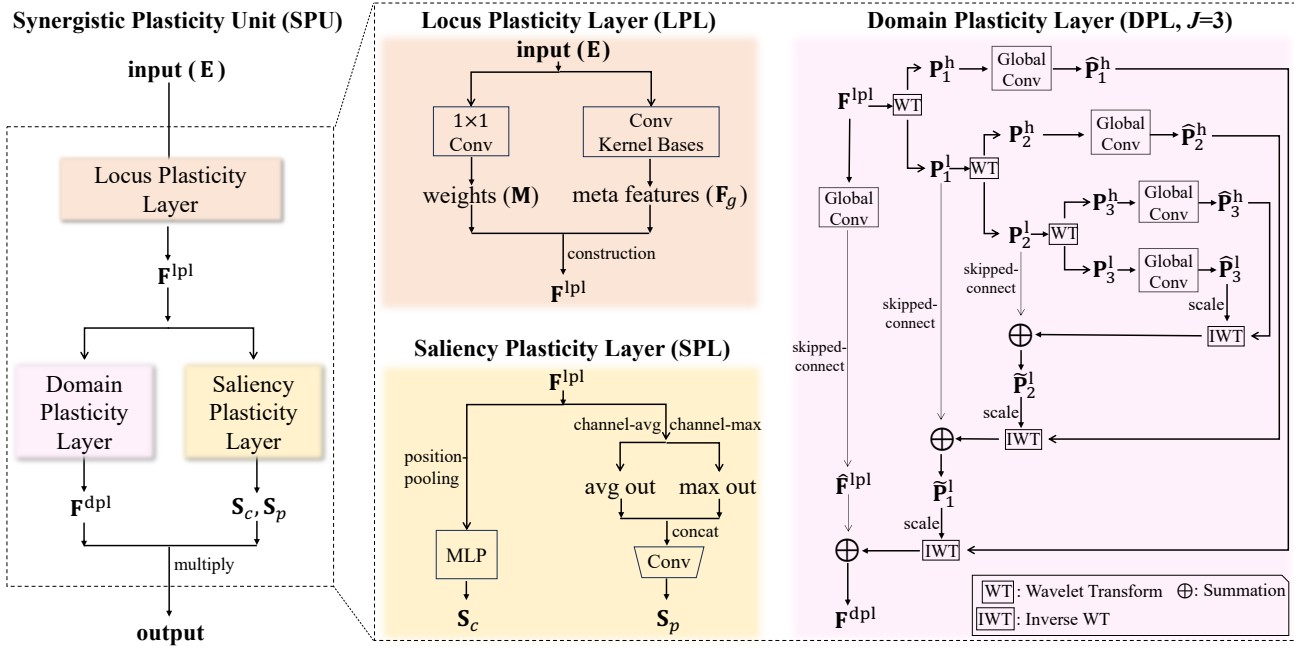

Figure 2. **The architecture of the SPU, which consists of the LPL, the DPL, and the SPL.** The LPL extracts features at each location by combining input-aware, token-specific coefficients with meta-features generated by the conv kernel bases. The DPL models global and cross-domain patterns by utilizing a global conv and the wavelet transform. Finally, the SPL generates dynamic saliency scores along channel and position axes to amplify critical information and suppress noise.

$0, \ldots, T - 1$. These are basic features used for token-level feature construction. Concurrently, a weight generation branch uses a 1×1 convolution on the input embedding $\mathbf{E}$ to yield a matrix of token-specific weights, $\mathbf{M} \in \mathbb{R}^{L \times T}$. The LPL's output of the $j$-th token, $\mathbf{F}_j^{\mathrm{lpl}}$, is then constructed by using the weights $\mathbf{M}_j = [m_{j,0}, m_{j,1}, \ldots, m_{j,T-1}]$ to create a weighted sum of the corresponding feature vectors $\mathbf{F}_{i,j}$ from the meta features:

$$\mathbf{F}_j^{\mathrm{lpl}} = \sum_{i=0}^{T-1} m_{j,i} \cdot \mathbf{F}_{i,j}, \tag{1}$$

where $\mathbf{F}^{\mathrm{lpl}} \in \mathbb{R}^{B \times L \times D}$. This token-specific construction mechanism provides high flexibility, allowing local context to more effectively guide feature extraction.

### 3.1.2. DOMAIN PLASTICITY LAYER

Following the LPL, the Domain Plasticity Layer (DPL) receives the contextually-rich feature map $\mathbf{F}^{\mathrm{lpl}}$. The primary objective of the DPL is to capture features across different perspectives by concurrently modeling features in *both the time and frequency domains*, which is achieved by a complementary architecture composed of a Convolution Path and a Wavelet Path (Fig. 2(right)).

**Convolution Path** This path acts as a direct connection to capture global features in the time domain. The input $\mathbf{F}^{\mathrm{lpl}}$ is processed by a Global Convolution operator, yielding

the output $\hat{\mathbf{F}}^{\mathrm{lpl}}$. This operator, denoted as GC, performs a non-causal, input-dependent long convolution accelerated by the Fast Fourier Transform (FFT) (Nussbaumer, 1981). The details are provided in Alg. 1.

**Wavelet Path** This path performs a multi-resolution analysis of the input feature using a discrete Wavelet Transform (WT) (Bentley & McDonnell, 1994). The process involves three main stages: decomposition, frequency-specific processing, and reconstruction. First, the input $\mathbf{F}^{\mathrm{lpl}}$ is iteratively decomposed over $J$ wavelet levels as illustrated for $J = 3$ in Fig. 2(right). At each level $i \in [1, J]$, the low-frequency components from the previous level, $\mathbf{P}_{i-1}^{\mathrm{l}}$ (with $\mathbf{P}_0^{\mathrm{l}} = \mathbf{F}^{\mathrm{lpl}}$), are passed through the WT to produce new, lower-resolution components $\mathbf{P}_i^{\mathrm{l}}$ and a set of high-frequency components $\mathbf{P}_i^{\mathrm{h}}$:

$$(\mathbf{P}_i^{\mathrm{l}}, \mathbf{P}_i^{\mathrm{h}}) = \mathrm{WT}(\mathbf{P}_{i-1}^{\mathrm{l}}), \quad i = 1, \ldots, J. \tag{2}$$

Then, each set of high-frequency components $\mathbf{P}_i^{\mathrm{h}}$ and the final low-frequency components $\mathbf{P}_J^{\mathrm{l}}$ are processed by the GC operator. This allows the model to learn distinct patterns within different frequency bands:

$$\hat{\mathbf{P}}_i^{\mathrm{h}} = \mathrm{GC}(\mathbf{P}_i^{\mathrm{h}}), \quad i = 1, \ldots, J, \tag{3}$$

$$\hat{\mathbf{P}}_J^{\mathrm{l}} = \mathrm{GC}(\mathbf{P}_J^{\mathrm{l}}). \tag{4}$$

Finally, the signal is reconstructed iteratively, starting from the processed lowest-frequency components, $\hat{\mathbf{P}}_J^{\mathrm{l}}$. This multi-scale feature fusion is defined by the following re-

**Algorithm 1** Global Convolution operator (GC)

**Input:** Input sequence $\mathbf{F}^{\text{lpl}} \in \mathbb{R}^{B \times L \times D}$.

1: **Parameters:**
2:     Learnable decay rates $\delta \in \mathbb{R}^D$ and shift $s \in \mathbb{R}^1$.
3:     Adaptive Layer Normalization (AdaLN), conditioned by $\mathbf{F}^{\text{lpl}}$.

**Output:** Output sequence $\hat{\mathbf{F}}^{\text{lpl}} \in \mathbb{R}^{B \times L \times D}$.

4: *Generate symmetric time embedding $t$ to enforce kernel symmetry:*
5: $t_j \leftarrow \frac{2}{L-1} \left| j - \frac{L-1}{2} \right| \quad \forall j \in \{0, \dots, L-1\}$
6: *Generate complex positional embedding $z$:*
7: $z_{j,d} \leftarrow \exp(-\mathrm{i} \cdot \frac{2\pi f_d}{L} \cdot j)$   ▷ i is the imaginary unit, $f_d$ is predefined for the $d$-th dimension.
8: *Generate data-dependent base filter $h$:*
9: $h \leftarrow \text{MLP}(\text{AdaLN}(z \mid \mathbf{F}^{\text{lpl}}))$  ▷ $\mathbf{F}^{\text{lpl}}$ is aggregated and mapped to parameters for AdaLN modulation.
10: *Apply symmetric modulation via $t$ to create the symmetric (non-causal) kernel $p$:*
11: $p \leftarrow h \odot (\exp(-t \otimes |\delta|) + s)$
12: $\hat{\mathbf{F}}^{\text{lpl}} \leftarrow \text{iFFT}(\text{FFT}(\mathbf{F}^{\text{lpl}}) \odot \text{FFT}(p))$   ▷ $\odot$ denotes element-wise multiplication.
13: *Crop to correct the circular shift and align the input and output:*
14: $\hat{\mathbf{F}}^{\text{lpl}} \leftarrow \text{crop}(\hat{\mathbf{F}}^{\text{lpl}}, \text{start} = \lfloor L/2 \rfloor, \text{length} = L)$

15: **return** $\hat{\mathbf{F}}^{\text{lpl}}$

---

cursive formula, applied at each level $i$ from $J-1$ to 1:

$$\tilde{\mathbf{P}}_i^{\text{l}} = \mathbf{P}_i^{\text{l}} + \text{IWT}\left(\gamma_i \cdot \tilde{\mathbf{P}}_{i+1}^{\text{l}}, \quad \hat{\mathbf{P}}_{i+1}^{\text{h}}\right). \tag{5}$$

The recursion is initialized with the base case $\tilde{\mathbf{P}}_J^{\text{l}} = \hat{\mathbf{P}}_J^{\text{l}}$. At each step, the reconstructed signal from the level below ($\tilde{\mathbf{P}}_{i+1}^l$) is scaled by a learnable parameter $\gamma_i$ and then upsampled via the Inverse WT (IWT) using the processed high-frequency components ($\hat{\mathbf{P}}_{i+1}^{\text{h}}$). This result is then added to the original, unprocessed low-frequency components from the decomposition stage ($\mathbf{P}_i^{\text{l}}$) to form the final representations for the current, higher-resolution level, $\tilde{\mathbf{P}}_i^{\text{l}}$. This skip connection preserves the original low-frequency information, preventing the loss of critical patterns. After $J-1$ iterations of Eq. 5, the reconstructed signal $\tilde{\mathbf{P}}_1^{\text{l}}$ is obtained. The final reconstruction step, which combines time- and frequency-sensitive representations, is formulated as: $\mathbf{F}^{\text{dpl}} = \hat{\mathbf{F}}^{\text{lpl}} + \text{IWT}\left(\gamma_0 \cdot \tilde{\mathbf{P}}_1^{\text{l}}, \ \hat{\mathbf{P}}_1^{\text{h}}\right)$.

### 3.1.3. SALIENCY PLASTICITY LAYER

Simultaneously, the Saliency Plasticity Layer (SPL) receives the output from the LPL $\mathbf{F}^{\text{lpl}} \in \mathbb{R}^{B \times L \times D}$ as its input. The objective of the SPL is to determine the significance of the captured features by generating dynamic, data-dependent weights across both channel and position axes. As illus-

---

trated in Fig. 2(right), the SPL is composed of two parallel branches: a Multi-Layer Perceptron (MLP) branch for channel saliency and a depthwise separable convolution branch for position saliency. In the first branch, the input $\mathbf{F}^{\text{lpl}}$ is aggregated along the position axis using average pooling to create a channel descriptor vector. This vector is then processed by an MLP to yield the channel saliency scores, $\mathbf{S}_c \in \mathbb{R}^{B \times 1 \times D}$. In parallel, the second branch computes features along the channel axis by calculating both the average and the maximum values of $\mathbf{F}^{\text{lpl}}$, which provides a richer position descriptor, enabling the model to capture both sharp motifs via peak activations and broader regulatory regions via average activations. These two values are concatenated and passed through a depthwise separable convolution to perform spatial integration and produce the position saliency scores, $\mathbf{S}_p \in \mathbb{R}^{B \times L \times 1}$. Finally, $\mathbf{S}_c$ and $\mathbf{S}_p$ are broadcast to match the dimensions of the features from the DPL, $\mathbf{F}^{\text{dpl}}$, and are multiplied element-wise. This operation dynamically refines the features to produce the final output of the SPU. By explicitly modeling both channel-wise ('what') and positional ('where') saliency, SPL provides a more comprehensive feature refinement mechanism than approaches that are solely channel-focused or locally gated.

### 3.2. Theoretical Analysis

We motivate SPU's architecture as a joint design driven by both DNA biological priors and signal processing constraints. Specifically, we analyze its structural requirements through the Time-Frequency Uncertainty Principle (TUP) (Gabor, 1946). While DNA sequences are discrete, we model them as samples from continuous signals to derive these theoretical bounds. Here, $\sigma_t$ and $\sigma_\omega$ represent the signal's spread in the time (sequential) and frequency domains, respectively.

**Lemma 3.1** (Static Resolution Limitation)**.** *Existing architectures are constrained by fixed time-frequency tiling strategies. For a fixed kernel size $k$ (CNNs) or a global sequence length $L$ (FFT/Hyena), the receptive field implies:*

$$\begin{aligned} \mathcal{R}_{CNN} &\implies (\sigma_t \sim \mathcal{O}(k), \quad \sigma_\omega \sim \mathcal{O}(1/k)) \\ \mathcal{R}_{FFT} &\implies (\sigma_t \sim \mathcal{O}(L), \quad \sigma_\omega \sim \mathcal{O}(1/L)). \end{aligned} \tag{6}$$

*Proof.* See Appendix A.1. $\mathcal{O}(\cdot)$ denotes the standard Big-O notation for asymptotic scaling. This limitation implies that CNNs excel at capturing transient motifs but miss global periodicity (due to coarse frequency resolution), while global mixing models capture global features but may blur local motifs (due to lack of time localization).

**Theorem 3.2** (Logarithmic Multi-Resolution Tiling)**.** *The SPU architecture, via the Wavelet Path in DPL, explicitly constructs a multiresolution analysis (MRA) structure. The effective time-frequency window $\mathcal{W}_i \sim \sigma_t^{(i)} \times \sigma_\omega^{(i)}$ at de-*

*Table 1.* **Performance comparison on Genomic Benchmark.** Top-1 accuracy (%, ↑) is reported for the officially released pre-trained models. 'ConvNova' is pre-trained using the officially released code.

|  | HyenaDNA (436k) | CADUCEUS-PH (470k) | ConvNova (386k) | xLSTM-PH (500k) | SPU-DNA (ours) (490k) |
|---|---|---|---|---|---|
| Mouse Enhancers | 78.03±0.19 | 78.60±1.32 | 79.17±0.87 | 78.48±0.94 | **82.38±0.66** |
| Coding vs. Intergenomic | 90.43±0.05 | 91.86±0.28 | 93.35±1.52 | 93.11±1.38 | **94.46±0.47** |
| Human vs. Worm | 95.83±0.16 | 96.67±0.13 | 96.85±0.85 | 96.71±0.65 | **97.19±0.75** |
| Human Enhancers Cohn | 72.46±0.21 | 73.61±0.47 | 73.82±0.56 | 73.88±0.56 | **74.96±0.35** |
| Human Enhancers Ensembl | 89.43±0.02 | 90.11±0.32 | 90.10±0.16 | 90.42±1.04 | **91.08±0.61** |
| Human Regulatory | 88.45±0.08 | 87.82±0.46 | 88.21±1.03 | 87.67±0.31 | **94.20±0.23** |
| Human Nontata Promoters | 94.74±0.20 | 94.72±0.48 | **95.21±1.26** | 95.11±0.86 | 94.78±0.51 |
| Human OCR Ensembl | 78.14±2.01 | 80.17±0.15 | 79.61±0.42 | 81.72±1.16 | **81.95±1.01** |

*composition level $i$ satisfies the scaling law:*

$$\sigma_t^{(i)} \propto 2^i \sigma_t^{(0)}, \quad \sigma_\omega^{(i)} \propto 2^{-i} \sigma_\omega^{(0)} \quad (7)$$

*for all $i \in \{1, \ldots, J\}$. Consequently, SPU covers the signal space with a union of hierarchically scaled Heisenberg boxes $\bigcup_i \mathcal{W}_i$, allowing the learned Global Convolutions to process features within optimal locality constraints.*

*Proof.* See Appendix A.2. SPU logarithmically tiles the time-frequency plane, with LPL acting as a pre-conditioner at the sub-band limit to establish the initial basis $\mathbf{F}^{\text{lpl}}$. This synergy provides a more comprehensive scale coverage from sharp local motifs to long-range (large $i$) dependencies.

### 3.3. SPU-DNA model

**Computational Complexity** Given an input sequence of length $L$, the LPL (convs) and the SPL (MLP and convs)

both operate with linear complexity, $\mathcal{O}(L)$. The complexity of the DPL is determined by its wavelet analysis ($\mathcal{O}(L)$) and FFT-based global conv ($\mathcal{O}(L\log_2 L)$). Therefore, the global conv is the computational bottleneck, making the overall complexity of the SPU quasi-linear at $\mathcal{O}(L\log_2 L)$.

**Constructing DNA models with SPU** The SPU is scalable for constructing DNA models of varying sizes, termed SPU-DNA. Specifically, the SPU-DNA first embeds a tokenized DNA sequence into an embedding, $\mathbf{E}$, which is then processed by an encoder—a stack of SPU layers, each configurable as 'Pre-Norm' or 'Post-Norm', similar to standard Transformer blocks. The features are then passed to a task-specific head for prediction (see B for further details).

**Pre-training and Fine-tuning** Following prior works (Nguyen et al., 2023; Bo et al., 2025), we pre-train our models on the human reference genome (hg38) using Masked Language Modeling (MLM). We apply a

*Table 2.* **Performance comparison on Nucleotide Transformer Benchmark.** Performance (%, ↑) is reported for the officially released pre-trained models. 'ConvNova' is pre-trained using the officially released code. Metrics vary by task: MCC for histone markers and enhancers, F1-score for promoters and splice site acceptor/donor, and accuracy for splice site all. The best results for models under 5M parameters are underlined, and the best overall results are bolded.

|  | >100M Param. | | | <5M Param. | | | | |
|---|---|---|---|---|---|---|---|---|
|  | DNABERT-V2 (117M) | NT-V2 (500M) | SPU-DNA (ours) (500M) | HyenaDNA (1.6M) | CADUCEUS-PH (1.9M) | ConvNova (1.7M) | xLSTM-PH (2.0M) | SPU-DNA (ours) (1.9M) |
| *Regulatory Annotation* | | | | | | | | |
| Promoter all | 97.06±0.11 | 97.61±0.12 | **98.35±0.13** | 96.22±0.29 | 96.13±0.10 | 96.83±0.21 | 96.80±0.76 | _97.86±0.11_ |
| Non-TATA | 97.18±0.16 | 97.53±0.14 | **98.61±0.05** | 95.60±0.25 | 96.03±0.16 | 96.19±0.53 | 96.11±0.62 | _97.12±0.10_ |
| TATA | 95.50±0.39 | 96.35±1.06 | **99.20±0.08** | 95.48±0.24 | 95.46±0.36 | 96.50±0.67 | 95.41±0.58 | _97.08±0.18_ |
| Enhancer | 55.81±1.60 | 56.62±3.20 | **61.35±0.21** | 53.82±0.40 | 50.68±2.41 | 57.78±0.29 | 55.01±0.33 | _58.81±0.21_ |
| Enhancer types | 43.78±3.01 | 45.95±2.30 | **62.98±0.12** | 48.49±0.85 | 39.21±1.85 | 49.89±0.60 | 47.01±0.52 | _51.17±0.12_ |
| *Splice Site Annotation* | | | | | | | | |
| Acceptor | 96.46±0.35 | 98.64±0.22 | **99.14±0.06** | 96.39±0.74 | 96.54±0.28 | 96.31±0.47 | 95.54±1.11 | _97.23±0.08_ |
| Donor | 96.11±0.23 | 98.86±0.18 | **99.30±0.17** | 95.26±0.63 | 94.68±0.92 | 96.80±0.55 | 95.69±1.05 | _97.86±0.17_ |
| All | 93.70±0.40 | 98.37±0.14 | **99.69±0.22** | 95.52±0.97 | 95.13±0.32 | 96.36±0.19 | 95.90±0.58 | _97.26±0.20_ |
| *Histone Markers* | | | | | | | | |
| H3 | 76.38±2.73 | 79.14±1.21 | **85.29±0.05** | 79.89±0.61 | 78.45±0.67 | 81.62±1.23 | 80.97±0.59 | _83.56±0.16_ |
| H3K4me1 | 50.75±1.20 | 54.21±0.74 | **59.53±0.12** | 51.81±0.79 | 47.20±0.89 | 53.87±1.38 | 52.66±1.25 | _55.88±0.26_ |
| H3K4me2 | 31.12±2.63 | 32.12±2.99 | **60.22±0.17** | 46.87±1.08 | 45.67±1.35 | 53.39±0.65 | 51.29±0.81 | _54.19±0.03_ |
| H3K4me3 | 36.59±1.74 | 40.25±3.09 | **66.98±0.11** | 49.29±0.62 | 47.92±1.67 | _58.21±1.59_ | 53.80±0.66 | 54.91±0.28 |
| H3K9ac | 53.68±0.90 | 56.62±0.88 | **68.23±0.19** | 59.02±0.54 | 60.39±0.92 | 64.12±0.87 | 62.91±0.37 | _65.56±0.17_ |
| H3K14ac | 50.20±1.14 | 54.07±1.67 | **71.37±0.14** | 55.70±0.39 | 58.54±1.01 | 59.47±0.96 | 59.25±0.42 | _62.39±0.22_ |
| H3K36me3 | 59.24±0.88 | 62.74±0.93 | **69.38±0.23** | 61.35±0.91 | 59.99±1.03 | 62.98±0.39 | 62.61±1.66 | _64.66±0.12_ |
| H3K79me3 | 60.48±1.08 | 63.71±1.02 | **73.06±0.11** | 67.51±0.51 | 67.03±0.62 | 70.28±1.11 | 69.88±0.47 | _70.98±0.65_ |
| H4 | 79.27±0.54 | 79.66±0.63 | **85.96±0.12** | 77.52±0.77 | 78.21±0.71 | 81.31±0.45 | 81.35±0.47 | _83.62±0.36_ |
| H4ac | 46.17±2.59 | 49.52±1.62 | **66.95±0.25** | 56.46±1.11 | 57.09±1.68 | _62.10±0.99_ | 59.54±0.86 | 56.98±0.21 |

*Table 3.* **Performance (%, ↑) comparison on Chromatin Profile Prediction.**

|  | TF | DHS | HM |
|---|---|---|---|
| DNABERT-V2 (117M) | 96.16 | 92.50 | 86.18 |
| NT-V2 (500M) | 96.76 | 93.39 | 86.76 |
| HyenaDNA (6.5M) | 95.91 | 92.33 | 85.46 |
| ConvNova (27.4M) | 96.99 | 93.71 | 86.70 |
| SPU-DNA (6.5M) | **97.25** | **94.01** | **87.02** |

15% random masking ratio to learn bidirectional contextual dependencies. For fine-tuning, we append task-specific heads to the encoder output. Specifically, for sequence classification, the hidden states are aggregated via average pooling into a global vector, which is then processed by a linear projection layer to generate final results.

# 4. Experiments

In this section, we evaluate our model's performance on tasks involving both short- and long-range DNA sequences. We then delve into the effectiveness and efficiency of SPU through ablation studies and in-depth model analyses.

## 4.1. Short Range Sequence Modeling

**Genomic Benchmark (Grešová et al., 2023)** This benchmark involves a classification task on 8 datasets, we report the mean and standard deviation of accuracy over multiple runs with different random seeds. As shown in Table 1, SPU-DNA outperforms the state-of-the-art on 7 of the 8 datasets, achieving absolute improvements of up to 5.75%.

**Nucleotide Transformer Benchmark (Dalla-Torre et al., 2025)** It contains 18 datasets across histone marker prediction, regulatory annotation, and splice site annotation. For each, we report the mean accuracy and standard deviation over multiple runs with different random seeds. The results in Table 2 highlight the effectiveness of our method. In the <5M parameter regime, our model achieves the best performance on 16 of the 18 datasets. When scaled to exceed 100M parameters, our approach leads on all the datasets. The significant performance gains observed with increased model size further validate the scalability of our approach.

## 4.2. Long Range Sequence Modeling

**Chromatin Profile Prediction** This multi-label classification task aims to predict chromatin profiles and epigenetic markers. We use the dataset from DeepSEA (Zhou & Troyanskaya, 2015), which includes 919 binary labels across transcription factor (TF) binding, DNase I-hypersensitive sites (DHS), and histone mark (HM) profiles. Following previous works (Nguyen et al., 2023; Bo et al., 2025), we measure performance using the AUROC score. As shown in Table 3, our model surpasses state-of-the-art methods in

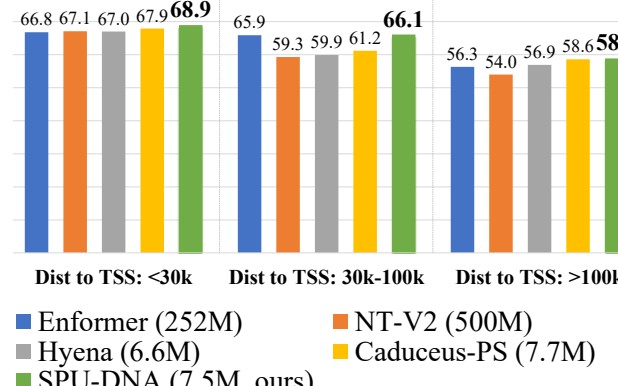

*Figure 3.* **Comparison on variant effects prediction.** AUROC (%, ↑) is reported for varying distances to the nearest Transcription Start Site (TSS): <30k, 30k-100k, >100k.

*Table 4.* **Ablation of the plasticity layers.**

|  | w/o LPL | w/o DPL | w/o SPL | SPU |
|---|---|---|---|---|
| Mouse Enhancers | 79.39 | 79.21 | 81.02 | **82.38** |
| Coding vs. Intergenomic | 91.06 | 89.98 | 92.11 | **94.46** |
| Human vs. Worm | 94.67 | 90.01 | 95.22 | **97.19** |
| Human Enhancers Cohn | 71.19 | 69.53 | 72.01 | **74.96** |
| Human Enhancers Ensembl | 88.66 | 85.91 | 90.29 | **91.08** |
| Human Regulatory | 90.34 | 89.19 | 91.36 | **94.20** |
| Human Nontata Promoters | 91.05 | 88.35 | 93.19 | **94.78** |
| Human OCR Ensembl | 78.61 | 76.20 | 79.38 | **81.95** |

all profiles while having the smallest parameter count.

**Variant Effect Prediction** This task evaluates the model's ability to predict the impact of SNPs on gene expression over long sequences. Consistent with prior work (Schiff et al., 2024), all other methods use an input **sequence length of 131k**, while NT and Enformer use lengths of 12k and 196k, respectively. The average results over multiple runs with different random seeds are reported. As shown in Fig. 3, our method outperforms competing approaches in all evaluated settings. This strong performance is achieved with a relatively small parameter count, highlighting the model's

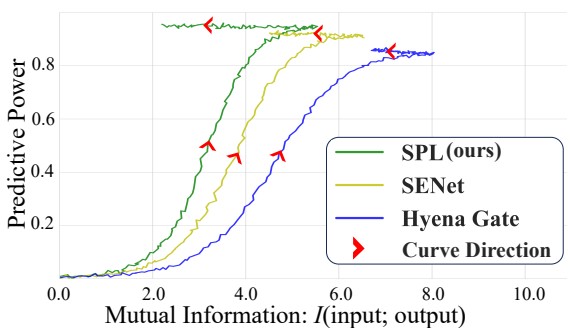

*Figure 4.* **Comparison of IB curves.** The curve direction shows the evolution of the x- and y-axis metrics during the training.

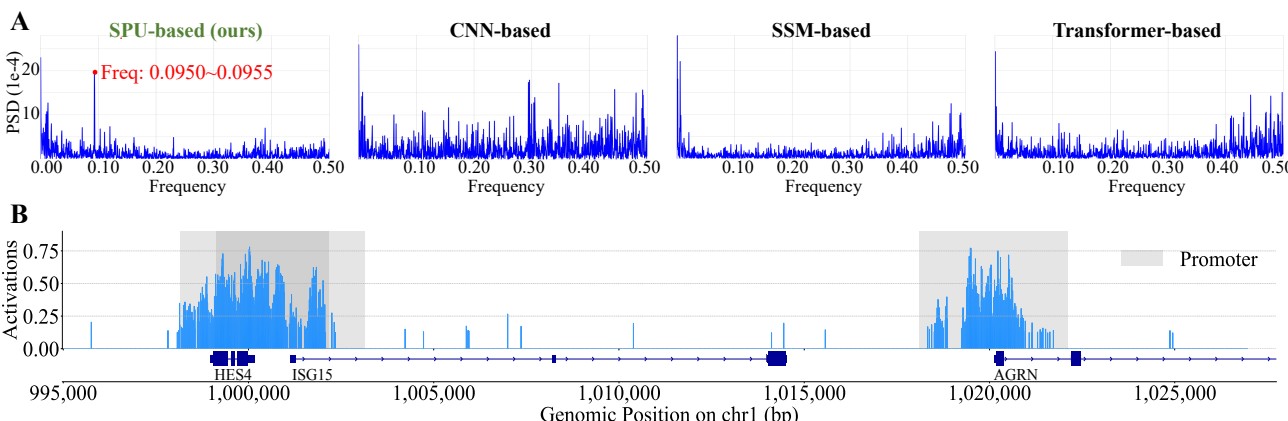

*Figure 5.* (A). **Power spectrum analysis** of periodic sequences processed by SPU-based and other models. (B). **Visualization of activations (blue) of a disentangled feature and promoter regions (gray).** Below are the IGV-style annotations for corresponding genes: HES4(-), ISG15(+), and ARGN(+). Arrows indicate transcription direction, with rectangles for CDS (thick) and UTR (thin).

effectiveness in capturing long-range dependencies.

### 4.3. Ablation Study and In-depth Analysis

**Plasticity Layers.** We conduct ablations on LPL, DPL, and SPL separately to validate their contributions. As shown in Table 4, removing any plasticity layer leads to a notable decline in performance. This confirms that the LPL, DPL, and SPL modules are structurally and functionally complementary; their synergistic integration is essential for achieving optimal results. (See Appendices L and M for more results.)

**Salient Feature Refining** We evaluate SPL against SENet (Hu et al., 2018) and Hyena gating via the *Information Bottleneck (IB)* framework (Tishby et al., 2000). Fig. 4 illustrates the training dynamics over 200 epochs. SPL demonstrates a superior trajectory, simultaneously achieving the highest predictive power ($I(\text{output}; \text{label})$) and maximal information compression ($I(\text{input}; \text{output})$). This empirically validates SPL as an optimal feature refining mechanism that effectively filters noise while retaining critical predictive information.

**Model Efficiency** We benchmark efficiency against Mamba, Hyena, and Transformers ($\approx$12M parameters, 100k seq-length, A800 GPUs) in Table 5. Notably, our model achieves the lowest FLOPs among all methods. While currently trailing mature SSMs slightly in runtime due to the absence of custom CUDA kernels, our model outperforms Transformers in both memory and speed. Crucially, the lower theoretical FLOPs indicate substantial potential for further acceleration through engineering optimization, positioning our approach as a highly efficient alternative with superior accuracy-efficiency balance.

**Periodic Sequence Processing** We extract features from a nucleosome region(hg19, chr1:1005266-1007266), which is known to exhibit a dinucleotide periodicity of about 0.095

*Table 5.* **Efficiency comparison on 100k sequence length.**

| | FLOPs | VRAM | Wall Clock |
|---|---|---|---|
| *(a) Training - Forward + Backward* | | | |
| Mamba | 5,619 G | 5.8 GB | 0.46s |
| Hyena | 5,573 G | 6.2 GB | 0.89s |
| Standard Transformer | 517,328 G | *OOM* | N/A |
| Transformer + FlashAttn | 516,931 G | 7.8 GB | 4.87s |
| SPU (Ours) | 5,431 G | 6.8 GB | 0.93s |
| *(b) Inference - Forward Only* | | | |
| Mamba | 1,452 G | 0.5 GB | 0.13s |
| Hyena | 1,391 G | 1.1 GB | 0.26s |
| Standard Transformer | 125,392 G | 28.7 GB | 19.30s |
| Transformer + FlashAttn | 125,281 G | 2.4 GB | 1.30s |
| SPU (Ours) | 1,316 G | 1.2 GB | 0.29s |

cycles/bp (Reynolds et al., 2009). We then compute the Power Spectral Density (PSD) of the extracted features. As shown in Fig. 5(A), the SPU's features display a distinct dual peak at 0.0950 and 0.0955 cycles/bp, matching the specific periodicity. In contrast, the other models' outputs show no significant peak in this frequency band. This result highlights the sensitivity of our model to periodic signals in DNA, a direct benefit of its multi-domain modeling.

**Feature Interpretability Analysis** To probe the biological meaning of our model's pre-trained embeddings, we train a Sparse Autoencoder (SAE) (Bussmann et al., 2024) to disentangle them. We analyze the resulting sparse features for correlation with promoter regions ($\pm$2,000 bp around TSS). The analysis reveals multiple promoter-associated features, with feature channel 1990 being a prime example. As visualized for a specific locus in Fig. 5(B), the activation peaks of this feature show strong spatial alignment with the promoter regions, which confirms that our model autonomously identifies key regulatory elements like promoters from raw DNA sequences alone, without relying on

explicit genomic annotations (see K and P for more details).

## 5. Conclusion

In this work, we introduce the Synergistic Plasticity Unit (SPU) to transcend the limitations of static, single-domain models via multi-level plasticity. SPU organically integrates precise local dynamics (LPL), global cross-domain dependencies (DPL), and selective information flow (SPL), enabling dynamic adaptation to genomic complexity. Extensive evaluations demonstrate that SPU-DNA achieves state-of-the-art performance with high efficiency. Ultimately, this work establishes SPU as a theoretically grounded, empirically robust, and biologically interpretable paradigm, paving the way for the next generation of DNA foundation models.

## Acknowledgements

This work was supported by Damo Academy (Hupan Laboratory) [1].

## Impact Statement

This work aims to advance the intersection of machine learning and computational biology. All datasets utilized in this study are publicly available and anonymized, ensuring no concerns regarding personal privacy or ethical violations. While our proposed method holds theoretical and practical value for genomics and is expected to positively influence the research community, we anticipate these impacts to be constructive and aligned with ethical standards. Consequently, we assess that this work poses no foreseeable negative risks to society.

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

# A. Theoretical Proofs

To establish fundamental theoretical bounds using the Time-Frequency Uncertainty Principle (TUP) (Gabor, 1946), we require a continuous analytical framework. While the input DNA sequence is discrete, denoted as $\mathbf{x} \in \mathbb{R}^{L \times D}$, we postulate the existence of an underlying continuous signal $f(t) \in L^2(\mathbb{R})$ such that the discrete features are samples of $f(t)$ at integer indices (i.e., $x[n] = f(nT)$). This continuous relaxation allows us to leverage tools from harmonic analysis, and the fundamental insights about time-frequency localization trade-offs are grounded in the TUP, which provides principled justification for SPU's design.

We define the time center $t_c$ and variance $\sigma_t^2$ of a non-zero function $\phi(t) \in L^2(\mathbb{R})$ based on its energy density:

$$t_c = \frac{1}{\|\phi\|^2} \int_{-\infty}^{\infty} t|\phi(t)|^2 dt, \quad \sigma_t^2 = \frac{1}{\|\phi\|^2} \int_{-\infty}^{\infty} (t - t_c)^2 |\phi(t)|^2 dt \tag{8}$$

where $\|\phi\|^2 = \int |\phi(t)|^2 dt$ is the signal energy. Analogous definitions apply to the frequency variance $\sigma_\omega^2$ using the Fourier transform $\hat{\phi}(\omega)$. The TUP imposes the fundamental limit $\sigma_t \sigma_\omega \geq \frac{1}{2}$.

## A.1. Proof of Lemma 3.1 (Static Resolution Limitation)

**Case 1: Convolutional Neural Networks (CNNs).** Consider a standard CNN layer parameterized by a discrete kernel $g[n]$ of size $k$. In the continuous limit, this corresponds to a kernel function $g(t)$ with compact support, i.e., $g(t) = 0$ for $|t| > k/2$. Assuming the energy is uniformly distributed within the support for a worst-case analysis, the time spread is bounded by the kernel size:

$$\sigma_t^{CNN} \sim \mathcal{O}(k) \tag{9}$$

According to the scaling property of the Fourier transform, a signal confined to a small temporal duration $k$ must span a wide spectral bandwidth. Formally, by the TUP lower bound:

$$\sigma_\omega^{CNN} \geq \frac{1}{2\sigma_t^{CNN}} \sim \mathcal{O}\left(\frac{1}{k}\right) \tag{10}$$

In genomic sequence modeling, local kernels typically satisfy $k \ll L$ (where $L$ is the sequence length). Consequently, the frequency resolution $\Delta\omega \sim 1/k$ is significantly coarser than the fundamental frequency resolution of the sequence, $1/L$. **Implication:** A CNN kernel inherently acts as a broadband filter. It lacks the spectral resolution to discriminate fine-grained periodic patterns (e.g., specific codon periodicities) if their spectral width is narrower than $\mathcal{O}(1/k)$.

**Case 2: Global Mixing Models (e.g., Transformer, Fourier-Global Conv).** Standard global mixing mechanisms utilize basis functions (e.g., positional encodings $e^{i\omega t}$ in Transformers or global filters in FFT-based models) that span the entire sequence domain $L$.

$$\sigma_t^{Global} \sim \mathcal{O}(L), \quad \sigma_\omega^{Global} \sim \mathcal{O}\left(\frac{1}{L}\right) \tag{11}$$

While these models achieve maximal frequency resolution ($\sigma_\omega$ is minimized), the time uncertainty is maximized ($\sigma_t \to L$). **Implication:** For a high-frequency transient event (e.g., a localized TATA-box motif), the positional information is delocalized across the entire sequence length $L$. Recovering the precise location of such a motif requires the constructive interference of a vast number of global coefficients, which poses a significant optimization challenge.

## A.2. Proof of Theorem 3.2 (Logarithmic Multi-Resolution Tiling)

The SPU architecture circumvents the fixed trade-off in Lemma 3.1 by constructing a Multiresolution Analysis (MRA) structure via the Domain Plasticity Layer (DPL).

**Notations and Definitions.** Let $\psi(t)$ be the mother wavelet function used for the decomposition (analysis) process in the DPL. To support the theoretical derivations, we explicitly define the associated variants of $\psi$:

- **Fourier Transform ($\hat{\psi}$):** Let $\hat{\psi}(\omega) = \int_{-\infty}^{\infty} \psi(t)e^{-i\omega t} dt$ denote the Fourier transform of the wavelet.

- **Dual Wavelet ($\tilde{\psi}$):** Let $\tilde{\psi}(t)$ denote the dual (synthesis) wavelet associated with $\psi(t)$.

The DPL decomposes the input signal into hierarchical levels $i = 1, \ldots, J$. The effective basis functions at level $i$ are given by the dilated and translated wavelets:

$$\psi_{i,k}(t) = 2^{-i/2}\psi(2^{-i}t - k) \tag{12}$$

**1. Time Localization Derivation:** We evaluate the time spread $\sigma_t^{(i)}$ for the basis at level $i$. First, we note that the time center of the dilated wavelet scales and shifts as $t_c^{(i)} = 2^i t_c^{(0)} + 2^i k$. Using the change of variable $u = 2^{-i}t - k$ (implying $dt = 2^i du$ and $t = 2^i(u + k)$):

$$(\sigma_t^{(i)})^2 = \frac{1}{\|\psi_{i,k}\|^2} \int_{-\infty}^{\infty} (t - t_c^{(i)})^2 |2^{-i/2}\psi(2^{-i}t - k)|^2 dt \tag{13}$$

$$= \int_{-\infty}^{\infty} \left[ 2^i(u + k) - (2^i t_c^{(0)} + 2^i k) \right]^2 |\psi(u)|^2 du \quad \text{(substituting } t \text{ and } t_c^{(i)}) \tag{14}$$

$$= \int_{-\infty}^{\infty} (2^i(u - t_c^{(0)}))^2 |\psi(u)|^2 du \quad \text{(shift } k \text{ cancels out)} \tag{15}$$

$$= 2^{2i} \int_{-\infty}^{\infty} (u - t_c^{(0)})^2 |\psi(u)|^2 du \tag{16}$$

$$= 2^{2i}(\sigma_t^{(0)})^2 \tag{17}$$

Taking the square root yields the scaling law:

$$\sigma_t^{(i)} = 2^i \sigma_t^{(0)} \tag{18}$$

This demonstrates that the effective temporal receptive field grows exponentially with the level $i$.

**2. Frequency Localization Derivation:** By the properties of the Fourier transform, the spectrum of the dilated wavelet is $\hat{\psi}_{i,k}(\omega) = 2^{i/2}\hat{\psi}(2^i\omega)$. The frequency variance scales inversely:

$$\sigma_\omega^{(i)} = 2^{-i}\sigma_\omega^{(0)} \tag{19}$$

**3. Adaptive Tiling Mechanism:** The product $\sigma_t^{(i)}\sigma_\omega^{(i)}$ remains constant (bounded by TUP), but the tiling of the time-frequency plane changes adaptively:

$$\frac{\sigma_t^{(i)}}{\sigma_\omega^{(i)}} = 2^{2i}\frac{\sigma_t^{(0)}}{\sigma_\omega^{(0)}} \tag{20}$$

In the DPL, the Global Convolution (GC) operator is applied to the specific frequency sub-band $\mathbf{P}_i^{\mathrm{h}}$. Mathematically, the effective operation on the underlying continuous signal $f(t)$ can be modeled as:

$$y(t) = \sum_{i=1}^{J} ((f * \psi_i) * h_i) * \tilde{\psi}_i \tag{21}$$

where $h_i$ is the learnable global kernel for level $i$. Crucially, even though $h_i$ has global support, it convolves with a signal $(f * \psi_i)$ that is concentrated in a band. Consequently, the learned dependencies at level $i$ are constrained to the time-frequency window $\mathcal{W}_i = [\sigma_t^{(i)}, \sigma_\omega^{(i)}]$. This allows the SPU to:

- Resolve sharp, local motifs using small $i$ (high frequency, low $\sigma_t$).

- Capture long-range periodic structures using large $i$ (low frequency, low $\sigma_\omega$).

The total representational capacity is given by the union of these adaptive subspaces:

$$\mathcal{S}_{SPU} = \mathrm{span}\left( \bigcup_i \{\psi_i * h_i\} \right) \tag{22}$$

This confirms that SPU provides a relatively comprehensive coverage of the time-frequency plane, resolving the dilemma faced by static CNN or Transformer architectures.

## B. SPU-based DNA Models

As shown in Fig. 6, in the 'Pre-Norm' architecture, layer normalization is applied before each sub-layer. The input is first normalized to **E** and then processed by the SPU. After the first residual connection, the result is normalized again before being passed to the FFN, followed by a final residual connection. In the 'Post-Norm' architecture, layer normalization is applied after each residual connection. The input feature map **E** first passes through the SPU module. The output is then added to the input via a residual connection, and the result is normalized. This process is repeated for the subsequent FFN. A standard linear input/output projection is applied before or after the SPL/SPU; these common operations are omitted from the figure to emphasize the key aspects of the model architecture.

### SPU-based DNA Model

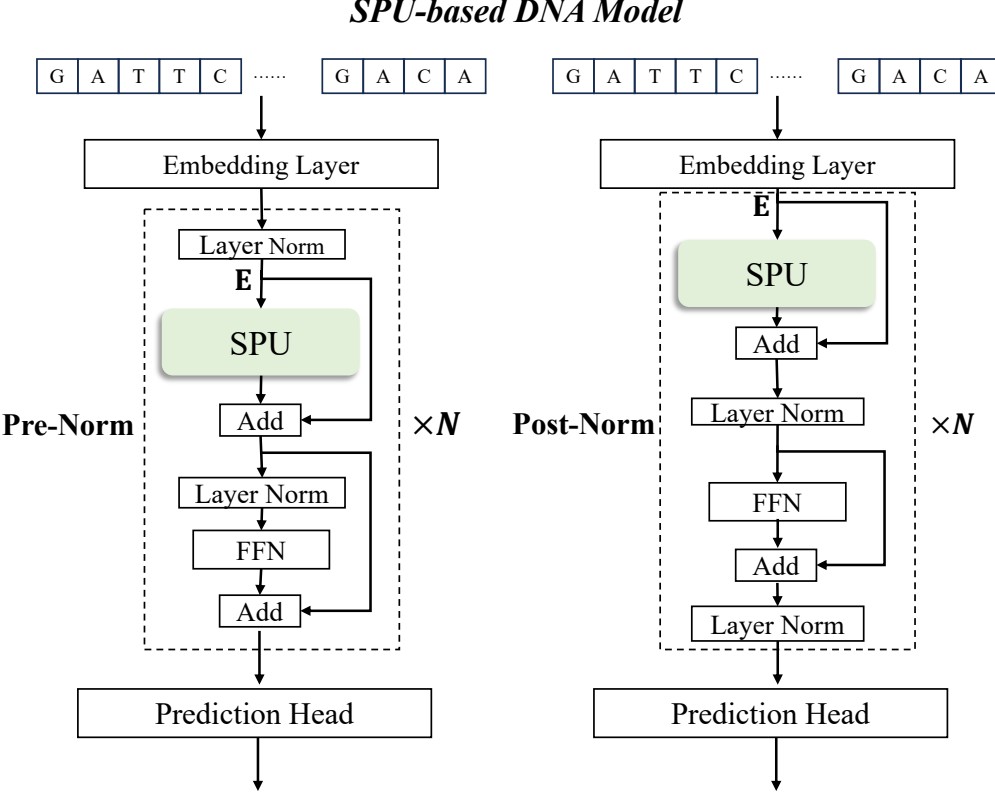

*Figure 6.* Architectural scaling of the SPU to create DNA models of different capacities.

## C. Details of Pre-training

During the pre-training phase, we trained the model for 10 epochs on the human genome (hg38) using the Adam optimizer, a batch size of 256, and a learning rate of 1e-4. We used a 1-mer tokenization ($k = 1$), where each nucleotide was treated as a single token. Within the LPL, we used convolutional kernel bases with sizes of [1, 3, 5, 7]. In the DPL, the wavelet transform was configured with 3 decomposition levels. In the SPL, the kernel size of the depthwise separable convolution was $7 \times 7$. All experiments were conducted on 8 NVIDIA A800 GPUs.

## D. Genomic Benchmark

For this benchmark, we constructed various-size DNA models using an encoder composed of SPUs stacked in a 'pre-norm' configuration. Consistent with prior work, the model was pre-trained on the human reference genome (hg38). For methods with multiple architectural variants (e.g., Caduceus and xLSTM), we selected their best-performing architecture for the specific task in our comparison.

## E. Nucleotide Transformer Benchmark

For this benchmark, we constructed various-size DNA models with an encoder composed SPUs stacked in a 'pre-norm' configuration. Consistent with prior work, the model was pre-trained on the human reference genome (hg38). During the fine-tuning phase, we adopted a batch size of 512 and a learning rate of 1e-3 for the $\approx 500M$ model, as our empirical results consistently indicated that the model achieves superior performance with larger batch sizes compared to smaller ones. For smaller models, we use a smaller learning rate and a smaller batch size. For methods with multiple architectural variants (e.g., Caduceus and xLSTM), we selected their best-performing architecture for the specific task in our comparison.

## F. Convolutional Kernel Bases

We conducted an ablation study on the Genomic Benchmark to assess the impact of varying the number of convolutional kernel bases, with the results detailed in Table 6. The findings indicate that performance improves with more kernel bases, but the gains diminish when four or more are used. This suggests that, for this dataset, four kernel bases of different sizes are adequate for capturing a comprehensive set of meta-features, leading to the construction of features with high discriminative capabilities.

*Table 6.* **Performance ($\%, \uparrow$) comparison of different sizes of 'conv kernel bases'.**

|  | [1] | [1, 3, 5] | [1, 2, 3] | [1, 3, 5, 7] | [1, 3, 4, 5, 7] |
|---|---|---|---|---|---|
| Mouse Enhancers | 80.01±0.63 | 81.93±0.49 | 81.12±0.52 | 82.38±0.66 | 81.92±0.77 |
| Human vs. Worm | 96.89±0.71 | 96.96±0.76 | 96.91±0.69 | 97.19±0.75 | 97.25±0.81 |
| Human Regulatory | 93.88±0.28 | 94.16±0.31 | 94.01±0.26 | 94.20±0.23 | 94.15±0.33 |
| Coding vs. Intergenomic | 93.86±0.41 | 94.35±0.47 | 94.43±0.58 | 94.46±0.47 | 94.59±0.71 |
| Human Enhancers Cohn | 74.93±0.32 | 74.96±0.29 | 74.81±0.41 | 74.96±0.35 | 74.85±0.27 |
| Human Enhancers Ensembl | 90.28±0.58 | 90.96±0.49 | 90.83±0.62 | 91.08±0.61 | 90.91±0.73 |
| Human Nontata Promoters | 93.12±0.46 | 94.39±0.53 | 94.82±0.61 | 94.78±0.51 | 94.10±0.50 |
| Human OCR Ensembl | 80.56±0.86 | 81.88±0.79 | 81.39±0.89 | 81.95±1.01 | 81.27±0.92 |

## G. Wavelet levels

We evaluated the model's performance across different numbers of wavelet decomposition levels, with the results presented in Table 7. The results indicate that, on the whole, performance improves with more levels, a benefit derived from the increasingly fine-grained frequency resolution. Nevertheless, these gains become marginal when the number of levels exceeds three. Therefore, to strike a balance between performance and computational efficiency, we set the number of wavelet decomposition levels to $J=3$ for our experiments.

*Table 7.* **Performance ($\%, \uparrow$) comparison of different wavelet levels $J$.**

|  | 1 | 2 | 3 | 4 | 5 |
|---|---|---|---|---|---|
| Mouse Enhancers | 81.11±0.64 | 81.66±0.69 | 82.38±0.66 | 81.91±0.78 | 81.95±0.53 |
| Human vs. Worm | 97.01±0.39 | 97.03±0.73 | 97.19±0.75 | 97.05±0.75 | 97.03±0.77 |
| Human Regulatory | 93.88±0.31 | 93.91±0.26 | 94.20±0.23 | 94.12±0.25 | 94.19±0.39 |
| Coding vs. Intergenomic | 94.41±0.47 | 94.39±0.44 | 94.46±0.47 | 94.46±0.51 | 94.46±0.46 |
| Human Enhancers Cohn | 74.81±0.32 | 74.93±0.41 | 74.96±0.35 | 74.95±0.38 | 74.93±0.29 |
| Human Enhancers Ensembl | 90.88±0.57 | 91.02±0.60 | 91.08±0.61 | 91.07±0.63 | 91.12±0.72 |
| Human Nontata Promoters | 93.99±0.55 | 94.26±0.83 | 94.78±0.51 | 94.79±0.78 | 94.76±0.58 |
| Human OCR Ensembl | 81.90±0.89 | 81.93±0.92 | 81.95±1.01 | 81.95±1.08 | 81.93±0.97 |

## H. Chromatin Profile Prediction

The prediction of epigenetic markers in non-coding regions is a fundamental task for elucidating the functional impact of disease-associated variants. For this work, we utilized the benchmark dataset established by DeepSEA (Zhou & Troyanskaya,

2015). We followed the official data partitioning scheme, which ensures a strict, non-overlapping split between training and testing sets based on chromosomes. The dataset comprises 919 chromatin features compiled from the ENCODE (ENCODE Project Consortium et al., 2012) and Roadmap (Kundaje et al., 2015), including 690 transcription factor binding profiles (TF), 125 DNase I-hypersensitive sites (DHS), and 104 histone mark (HM) profiles. Each sample consists of a 1000 bp DNA sequence. A binary label of 1 was assigned for a given feature if the central 200-bp region of the sequence overlapped by more than 50% with a corresponding peak region; otherwise, the label was 0. This process generated a 919-dimensional label vector for each sample, framing the problem as a large-scale multi-label classification task. As the original data were based on the hg19 reference genome, we used the LiftOver tool (Kent et al., 2002) to convert all genomic coordinates to hg38. Samples that failed the conversion or resulted in an inconsistent sequence length (approximately 0.5%) were discarded.

For this task, we constructed a DNA model using an encoder of 6 SPU layers with a feature dimension of 256. The model was trained with a batch size of 512 and a learning rate of 4e-4.

## I. Variant Effect Prediction

Our methodology for predicting the causal impact of SNPs on gene expression followed the established protocol from prior work (Schiff et al., 2024) and involved four key stages. First, SNPs with a SuSiE-derived (Wang et al., 2020a) causal probability greater than 0.9 (Avsec et al., 2021) were assigned a positive label for the task. Second, we structured the dataset by using chromosomes 9 and 10 as a held-out test set (Kao et al., 2024) and stratifying all data based on the SNP's distance to the nearest Transcription Start Site (TSS). Third, for feature extraction, we fed each SNP's full-length reference and alternative sequences (12k bp for Nucleotide Transformer, 196k bp for Enformer, and 131k bp for the other methods) into the accordingly pre-trained model (a 6-layer SPU encoder with a 256-dimensional feature space) to generate embeddings. These were then averaged within a 1536 bp window centered on both the reference and alternative SNP sequences, and concatenated with the tissue of origin. Finally, we trained a Support Vector Machine (SVM) with an RBF kernel on 5,000 random data points from each TSS distance stratum, evaluated performance using AUROC on the test set. The average results are reported over five independent runs. For methods with multiple architectural variants (e.g., Caduceus), we selected their best-performing architecture for the specific task in our comparison.

## J. Details of Periodic Sequence Processing

We performed feature extraction and Power Spectral Density (PSD) analysis on nucleosome-bound DNA sequences using ConvNova, Caduceus, and DNABERT-V2, which correspond to the CNN-based, SSM-based, and Transformer-based methods in Fig. 5, respectively.

## K. Details of Feature Interpretation

As mentioned previously, we sought to decompose SPU's pre-trained dense embeddings into more disentangled, meaningful biological features. To this end, we trained a BatchTopK Sparse Autoencoder (SAE) (Bussmann et al., 2024) to learn a disentangled sparse representation from our model's dense embeddings. The SAE was configured with a dictionary size of 8192 features and trained with an L1 regularization coefficient of 3e-4 to enforce sparsity. The training data for the SAE consisted of hidden state embeddings of 1 billion tokens extracted from the final layer of the SPU model.

We then systematically scanned the entire genome to identify which of the resulting SAE features were specifically associated with promoter regions. Here we defined promoter regions as the ±2,000 bp window around each Transcription Start Site (TSS) based on the GENCODE v48 annotation (Frankish et al., 2020). We employed a sliding window strategy with a stride of 8,192 bp to process the genome and fed the corresponding DNA sequence into the pre-trained SPU model to obtain its embedding, which was then passed through the trained SAE to compute the activation value for each of the 8192 features. This process yielded a genome-wide activation map for each feature. We specifically hypothesized that some features would be associated with promoter regions. To score the association for each feature, we computed the log-ratio of its activation values within promoter regions compared to non-promoter regions:

$$S_f = \log \left( \frac{\text{mean}(A_f^{\text{promoter}})}{\text{mean}(A_f^{\text{non-promoter}}) + \epsilon} \right) \tag{23}$$

where $A_f^{\text{promoter}}$ and $A_f^{\text{non-promoter}}$ are the collections of activation values for feature $f$ in promoter and non-promoter regions, respectively. A small constant $\epsilon$ was added for numerical stability. A high positive score $S_f$ indicates that the feature $f$ is a strong marker for promoters.

## L. Local Motif Extractors

As shown in Table 8, LPL's advantage as a local feature extractor is evident from its performance against other strong models(e.g., ConvNeXt (Liu et al., 2022), MambaOut (Yu & Wang, 2025), and GCB (Bo et al., 2025)). This advantage stems from its token-specific feature extraction, a mechanism that better leverages local context to accurately capture diverse local patterns and ultimately improve model performance.

*Table 8.* **Performance** $(\%, \uparrow)$ **comparison of local motif extractors.**

|  | Conv | CondConv (Yang et al., 2019) | ConvNeXt | MambaOut (Yu & Wang, 2025) | GCB (Bo et al., 2025) | LPL |
|---|---|---|---|---|---|---|
| Mouse Enhancers | 80.01±0.19 | 79.89±0.41 | 80.22±0.22 | 78.06±0.39 | 80.38±0.68 | **82.38±0.66** |
| Coding vs. Intergenomic | 92.33±0.09 | 92.11±0.18 | 92.38±0.31 | 93.19±0.23 | 93.23±0.14 | **94.46±0.47** |
| Human vs. Worm | 96.36±0.10 | 95.38±0.22 | 96.12±0.08 | 95.87±0.26 | 96.29±0.11 | **97.19±0.75** |
| Human Enhancers Cohn | 73.36±0.15 | 73.39±0.27 | 72.58±0.61 | 73.01±0.25 | 73.03±0.37 | **74.96±0.35** |
| Human Enhancers Ensembl | 89.78±0.06 | 89.66±0.12 | 87.85±0.44 | 88.03±0.38 | 87.29±0.25 | **91.08±0.61** |
| Human Regulatory | 91.12±0.27 | 92.37±0.32 | 91.99±0.39 | 92.96±0.78 | 91.20±0.71 | **94.20±0.23** |
| Human Nontata Promoters | 94.25±0.04 | 94.12±0.11 | 94.20±0.13 | 94.13±0.21 | 94.32±0.42 | **94.78±0.51** |
| Human OCR Ensembl | 80.01±0.26 | 79.95±0.31 | 80.81±0.21 | 80.35±0.26 | 80.11±0.28 | **81.95±1.01** |

## M. Global Dependency Capturing

We compare the DPL with other modules capable of capturing global dependencies, such as long convolution and self-attention. Table 9 shows that the DPL achieves the best performance. We posit that this stronger result arises because multi-domain features provide a distinct and complementary source of information, enabling the model to capture dependencies that are imperceptible to purely single-domain operations.

*Table 9.* **Performance** $(\%, \uparrow)$ **comparison of 'long conv', 'self-attention' and 'DPL'.**

|  | long conv | self attention | DPL |
|---|---|---|---|
| Promoter all | 95.82±0.26 | 96.11±0.21 | **97.86±0.11** |
| Non-TATA | 95.31±0.23 | 95.31±0.28 | **97.12±0.10** |
| TATA | 96.12±0.09 | 96.38±0.36 | **97.08±0.18** |
| Enhancer | 53.12±0.27 | 52.93±0.37 | **58.81±0.21** |
| Enhancer types | 48.72±0.61 | 47.86±0.39 | **51.17±0.12** |
| Splice Site Acceptor | 96.03±0.15 | 96.41±0.10 | **97.23±0.08** |
| Splice Site Donor | 96.46±0.24 | 96.85±0.31 | **97.86±0.17** |
| Splice Site All | 95.87±0.25 | 94.33±0.24 | **97.26±0.20** |
| H3 | 79.89±0.32 | 78.63±0.45 | **83.56±0.16** |
| H3K4me1 | 54.38±0.21 | 53.19±0.51 | **55.88±0.26** |
| H3K4me2 | 51.96±0.32 | 53.85±0.23 | **54.19±0.03** |
| H3K4me3 | 50.50±0.25 | 49.42±0.36 | **54.91±0.28** |
| H3K9ac | 60.93±0.26 | 59.49±0.61 | **65.56±0.17** |
| H3K14ac | 57.32±0.19 | 58.36±0.23 | **62.39±0.22** |
| H3K36me3 | 62.29±0.15 | 63.02±0.16 | **64.66±0.12** |
| H3K79me3 | 65.58±0.30 | 66.99±0.08 | **70.98±0.65** |
| H4 | 78.69±0.24 | 79.55±0.35 | **83.62±0.36** |
| H4ac | 56.57±0.12 | 55.92±0.06 | **56.98±0.21** |

## N. Scalability

To further demonstrate the scalability of our architecture, we scaled our model to 100M and 500M (0.5B) parameters and evaluated them on the NT dataset. As shown in Table 10, the results reveal a clear scaling law: the 100M model achieves substantial performance gains across all tasks compared to the 1.9M variant, and the 500M model yields further notable

*Table 10.* **Performance comparison on the Nucleotide Transformer Benchmark.** We compare our SPU-DNA models (scaled from 1.9M to 500M) against the NT-V2 (500M).

| | NT-V2 (500M) | SPU-DNA (1.9M) | SPU-DNA (100M) | SPU-DNA (500M) |
|---|---|---|---|---|
| Promoter all | 97.61±0.12 | 97.86±0.11 | 97.93±0.22 | **98.35±0.13** |
| Non-TATA | 97.53±0.14 | 97.12±0.10 | 97.96±0.11 | **98.61±0.05** |
| TATA | 96.35±1.06 | 97.08±0.18 | 98.11±0.17 | **99.20±0.08** |
| Enhancer | 56.62±3.20 | 58.81±0.21 | 60.02±0.22 | **61.35±0.21** |
| Enhancer types | 45.95±2.30 | 51.17±0.12 | 52.91±0.18 | **62.98±0.12** |
| Splice Site Acceptor | 98.64±0.22 | 97.23±0.08 | 98.26±0.15 | **99.14±0.06** |
| Splice Site Donor | 98.86±0.18 | 97.86±0.17 | 98.79±0.20 | **99.30±0.17** |
| Splice Site All | 98.37±0.14 | 97.26±0.20 | 98.61±0.35 | **99.69±0.22** |
| H3 | 79.14±1.21 | 83.56±0.16 | 84.21±0.16 | **85.29±0.05** |
| H3K4me1 | 54.21±0.74 | 55.88±0.26 | 57.21±0.34 | **59.53±0.12** |
| H3K4me2 | 32.12±2.99 | 54.19±0.03 | 57.45±0.26 | **60.22±0.17** |
| H3K4me3 | 40.25±3.09 | 54.91±0.28 | 60.93±0.28 | **66.98±0.11** |
| H3K9ac | 56.62±0.88 | 65.56±0.17 | 66.35±0.34 | **68.23±0.19** |
| H3K14ac | 54.07±1.67 | 62.39±0.22 | 65.46±0.29 | **71.37±0.14** |
| H3K36me3 | 62.74±0.93 | 64.66±0.12 | 66.66±0.42 | **69.38±0.23** |
| H3K79me3 | 63.71±1.02 | 70.98±0.65 | 69.65±0.44 | **73.06±0.11** |
| H4 | 79.66±0.63 | 83.62±0.36 | 84.99±0.52 | **85.96±0.12** |
| H4ac | 49.52±1.62 | 56.98±0.21 | 61.38±0.36 | **66.95±0.25** |

*Table 11.* **Performance comparison on the Gene Finding task (Bend dataset).** Metric: Matthews Correlation Coefficient (MCC).

| | NT-H (500M) | DNABERT-2 (117M) | GENA-LM (336M) | HyenaDNA (6.5M) | ConvNova (7.4M) | SPU-DNA (6.6M) |
|---|---|---|---|---|---|---|
| MCC | 0.41 | 0.43 | 0.52 | 0.35 | 0.55 | **0.68** |

improvements over the 100M model. Most notably, our 500M model outperforms NT-v2 at a comparable parameter scale. This finding strongly validates that our proposed architecture is not only highly effective but also scales more favorably than alternative genomic foundation models.

## O. Bend Gene Finding

To evaluate the model's capacity for complex genomic annotation, we conducted experiments on Gene finding (Marin et al., 2023). This task requires classifying nucleotides into functional roles (e.g., exons, introns, donors, acceptors) based on the GENCODE dataset (Harrow et al., 2012), which features sequences up to 14,000 bps in length. This presents a dual challenge: models must detect fine-grained local signals to identify exon-intron boundaries while also modeling long-range dependencies to maintain global structural consistency. The results, presented in Table 11, demonstrate that our model holds a clear and significant advantage over other state-of-the-art methods on this new benchmark as well.

## P. Feature Alignment

To evaluate the feature quality learned by different models, we pre-trained our SPU-DNA model (12.3M) alongside a Transformer (12.6M) and a Hyena model (12.8M), using hg38 DNA sequences with a unified 32k context length. For each model, we trained a corresponding BatchTopK SAE (Bussmann et al., 2024) with consistent hyperparameters to disentangle its representations and evaluated their alignment with genomic annotations (promoters, exons, and CDS regions). A genomic position was considered "activated" if the feature value was non-zero. To quantify performance, we calculated the F1 score for each feature and reported the average F1 score of the top-5 most relevant features for each genomic region type. As shown in Table 12, under this strictly controlled setting, our model captures genomic regulatory information more effectively than the compared methods. Furthermore, we have also provided the alignment of the features output by SPU-DNA with the CDS regions of the DNA sequence for reference, Fig. 7.

*Table 12.* **F1 Score between features and genomic regions.**

|          | Transformer | Hyena  | SPU    |
|----------|-------------|--------|--------|
| promoter | 0.0601      | 0.0900 | **0.1320** |
| exons    | 0.0758      | 0.0830 | **0.0984** |
| cds      | 0.0341      | 0.0500 | **0.0719** |

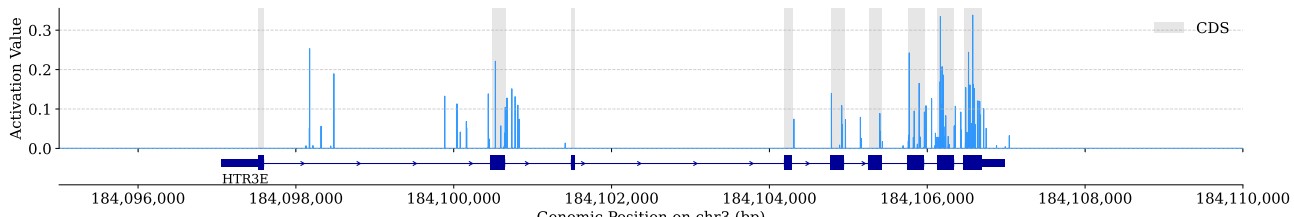

*Figure 7.* Visualization of feature alignment with cds regions.

## Q. Sequence Length of Pre-training

We pre-trained our model on the hg38 DNA sequence using two different sequence lengths: 1k and 12k (while maintaining the same model size). We then evaluated both pre-trained models on the Genomic Benchmark. The results are summarized below (Table 13).

The results clearly indicate that our model pre-trained on 12k sequences generally outperforms the model pre-trained on 1k sequences, either exceeding or matching the performance across multiple datasets.

We hypothesize that this improvement stems from the fact that pre-training on shorter sequences necessarily truncates long-range dependency information, which prevents the model from learning complete contextual associations. Utilizing a longer sequence length alleviates this issue, allowing the model to learn more accurate and holistic context-aware information. This outcome further validates that the "adaptive" nature of our architecture extends beyond mere feature pattern variability to encompass robustness and generalization across sequence lengths.

*Table 13.* **Ablation of pre-training length on Genomic Benchmark.**

|                          | SPU-DNA (1k length) | SPU-DNA (12k length) |
|--------------------------|---------------------|----------------------|
| Mouse Enhancers          | 82.38±0.66          | 84.15±0.71           |
| Coding vs. Intergenomic  | 94.46±0.47          | 94.49±0.42           |
| Human vs. Worm           | 97.19±0.75          | 97.16±0.68           |
| Human Enhancers Cohn     | 74.96±0.35          | 75.23±0.29           |
| Human Enhancers Ensembl  | 91.08±0.61          | 91.68±0.65           |
| Human Regulatory         | 94.20±0.23          | 94.26±0.31           |
| Human Nontata Promoters  | 94.78±0.51          | 94.98±0.16           |
| Human OCR Ensembl        | 81.95±1.01          | 83.01±0.95           |

## R. Periodic Signal Interference

To validate whether the multi-domain mechanism will introduce noise, we constructed contrastive groups (with vs. without periodic signals) derived from the Human Enhancers Cohn and Human Nontata Promoters datasets (Genomic Benchmark). In the groups containing periodic signals, we appended a 500-token random periodic signal (period = 5bp) to the end of each sequence sample. In the groups without periodic signals, we appended 500 non-periodic random bases. We then trained and evaluated the model on these datasets, where the objective remained enhancer or promoter prediction (independent of the appended signals). We evaluated both classification accuracy and the feature response ratio (defined as the ratio of feature activation intensity in the target region relative to the padding region), reporting the mean results over 5 independent runs. As shown in Table 14, the test accuracies on both groups are highly comparable. Crucially, for the group with periodic

signals, the feature response ratio is extremely high and aligns closely with that of the group without periodic signals. These results demonstrate that, for sequences either with or without periodic signals, our model effectively suppresses regions irrelevant to the prediction target rather than introducing multi-domain noise.

*Table 14.* **Comparison of sequences with or without periodic signal interference on `Human Enhancers Cohn` and `Human Nontata Promoters`.**

|  | Condition | Accuracy (%) | Feature Response Ratio |
|---|---|---|---|
| `Human Enhancers Cohn` | With periodic signal | 75.36±0.21 | 1218.03 |
|  | Without periodic signal | 75.29±0.19 | 1206.01 |
| `Human Nontata Promoters` | With periodic signal | 95.58±0.08 | 679.38 |
|  | Without periodic signal | 95.62±0.06 | 682.59 |

