# OpenReview forum: "Adaptive DNA Sequence Modeling via Synergistic Plasticity Units"
_ICML.cc/2026/Conference — ICML 2026 regular_

### Official Review · Reviewer_7GHL · 2026-03-08

**Soundness:** 2
**Presentation:** 2
**Significance:** 2
**Originality:** 2
**Overall Recommendation:** 3
**Confidence:** 4

**Summary:**

This paper proposes SPU, a comprehensive architecture for DNA modeling. Specifically, LPL uses multi-level convolutions and learnable weighting to learn DNA features. SPL applies a similar learnable weighting mechanism along both the sequence length and feature dimensions. DPL is relatively more novel conceptually: in order to model periodicity in DNA, it introduces wavelet transforms. SPU demonstrates strong performance on benchmarks such as NT Benchmark and Genomic Benchmark, and also provides a relatively novel showcase that the model is able to learn periodic patterns.

**Compliance With Llm Reviewing Policy:**

Affirmed.

**Final Justification:**

I appreciate the strong performance demonstrated by SPU within the evaluation scope of this paper. However, I am not convinced that SPU's relatively complex design motivations, such as long-context modeling and periodicity, provide actual benefits under the current benchmarks, as these benchmarks are primarily designed to evaluate motif detection on short sequences. In particular, the authors' comparatively low results on eQTL, combined with the reliance on outdated benchmarks, are not convincing. I do not think it is necessary for current work to strictly follow the experimental settings of papers from two or three years ago.

If the authors claim SPU to be a general-purpose foundational computational operator, I would encourage them to demonstrate its effectiveness in higher-impact domains, such as text.

In summary, I would like to maintain my score.

**Key Questions For Authors:**

1. Is the weighting coefficient passed through any nonlinearity?
2. Do the different global convolutions share parameters?

**Limitations:**

This paper does not include a limitation section. Please see Cons for suggestions.

**Strengths And Weaknesses:**

## Soundness

### Pros
1. The paper attempts to build a relatively complete architecture with a lot of reasonable designs.

### Cons
1. The benchmarks used are outdated. This paper appears to use the NT benchmark released at the preprint stage, but that benchmark has known construction issues. In the 2024 published NT paper, a revised version was already adopted.
2. Based on my checking, many of the reported results in this paper do not match those in previous work. Therefore, the authors should at least explain the experimental settings more clearly in the main paper to ensure fairness. For these benchmarks that do not have fixed data splits, selecting the best checkpoint can be tricky and can affect the comparison.

3. The ablation study only on Genomic Benchmark is not very informative. Moreover, such consistently large and clean ablation gains are themselves a bit surprising, especially since SPU does not seem to stably outperform baselines.

4. The eQTL results are too weak. Zero-shot results from models such as Enformer and AlphaGenome [1] can exceed 0.7. The generic performance on the important  task here makes me question the real utility of these complicated designs.

## Presentation

### Pros
1. The paper is structurally complete, with method, theory, and experiments all present.

### Cons
1. Figure 1 is not rigorous. Using the number of stars to represent the capability of different models is not a convincing.
2. Figure 2 is not very effective visually. It does not provide information that is easier to obtain than from the main text.

3. The module abbreviations are not very helpful. Names such as SPU, LPL, DPL, SPL etc., add cognitive overhead and are not easy to remember.

4. In Algorithm 1, line numbering starts before the actual algorithmic procedure begins.

## Significance

### Pros
1. Exploring periodicity and explicitly showcasing it could be meaningful.

### Cons
1. The tasks explored are outdated and of limited practical significance. The latest NT-v3 [2] has already abandoned the validation with these benchmarks. Genomic Benchmark and NT Benchmark are mainly designed to test motif detection ability, which has little connection to the authors' stated motivations around periodicity and long-context interaction.
## Originality

### Pros
1. Using wavelet transforms for modeling is conceptually novel in this context.

### Cons
1. LPL and SPL do not feel particularly novel to me. They look more like common techniques in deep learning architectures, with the overall idea being hierarchical modeling plus weighting mechanisms. Similar ideas have appeared in [3].

2. Periodicity modeling is somewhat novel, but [4] has already analyzed that CNNs themselves can learn periodicity, and models such as HyenaDNA also have some notion of periodicity modeling. In my view, the core difference here is replacing FFT with wavelets (disclaimer: I am not familiar with frequency-domain methods).

[1] Avsec, Žiga, et al. "AlphaGenome: advancing regulatory variant effect prediction with a unified DNA sequence model." BioRxiv (2025): 2025-06.

[2] Boshar, Sam, et al. "A foundational model for joint sequence-function multi-species modeling at scale for long-range genomic prediction." bioRxiv (2025): 2025-12.

[3] Bo, Yu, et al. "Revisiting convolution architecture in the realm of dna foundation models." arXiv preprint arXiv:2502.18538 (2025).

[4] Dudnyk, Kseniia, et al. "Sequence basis of transcription initiation in the human genome." Science 384.6694 (2024): eadj0116.

---

> ### Author Rebuttal · Authors · 2026-03-31
>
> Thank you for your thorough review and constructive suggestions. We address each point below.
> >Q1: Outdated benchmarks and task relevance.
>
> **R:** We acknowledge the new NT benchmark, but we use the original NT from the official repo since it remains widely adopted and ensures fair comparisons. All compared methods (including recent ones such as ConvNova and Bio-LSTM 2025) report and tune hyperparameters on this original version, and we reproduce them with the official code/settings. Switching to the new NT without official hyperparameters will confound the comparison.
> More broadly, our goal is to model short-range, long-range, and periodic DNA patterns (Lines 10-13); thus we evaluate GB/NT for short–medium sequences, CPP/VEP/BEND for long-range inputs up to 131 kb, and periodicity in Section 4.3.
>
> >Q2: Data splits & mismatched results.
>
> **R:** As noted in Tables 1–2, we reproduce prior methods using their public pre-trained checkpoints and official hyperparameters from their codebases, and evaluate all models under the same standardized train/test splits.
> Some mismatches with paper-reported numbers are also commonly observed in other works (e.g., Caduceus, ConvNova); our reproduced results (e.g., Hyena) are broadly consistent with those reported in these comparative works.
> All experiments in our work use a unified setup, run with multiple random seeds, and report mean performance for stable, reproducible results.
>
> >Q3: Ablation consistency.
>
> **R:** The ablations are expected because SPU’s layers are complementary and target different blind spots (Lemma 3.1). E.g., removing DPL drops frequency-domain modeling, while removing LPL weakens local motif capture, both causing clear degradation.
> These effects are not dataset-specific. Appendix M (Table 10) shows consistent drops across NT tasks when replacing DPL with other modules, indicating stable component synergy.
> We respectfully disagree that SPU does not stably outperform baselines: across our evaluations, SPU-DNA leads on most tasks, spanning short-range motifs, long-range dependencies, and mechanistic interpretability.
>
> >Q4: eQTL results.
>
> **R:** The statement that “zero-shot eQTL results from Enformer/AlphaGenome can exceed 0.7” is not directly comparable to our VEP results. Reasons:
> - Enformer/AlphaGenome setting: they are *sequence-to-function models* trained with **strong supervision** on functional genomics tracks, including expression assays. Their “zero-shot eQTL” is only “zero-shot” w.r.t. eQTL labels, not w.r.t. expression supervision, and some results use supervised downstream readouts (e.g., random forests).
> - Our VEP setting: variants are scored by a DNA model trained **without any functional-genomics supervision**, thus testing how well sequence-only modeling captures constraints correlated with regulatory impact.
>
> Moreover, we use a different dataset: our VEP follows a standardized protocol (by Caduceus), and under identical splits/code our method consistently outperforms other methods (including Enformer) across TSS-distance bins.
>
> >Q5: Presentation advice.
>
> **R:** We'll clarify that Fig. 1 is conceptual (not an empirical plot), and better introduce the abbreviations to reduce cognitive load. Fig. 2 gives the overall structure, with more details in the text. For Alg. 1, the formatting follows the official ICML template, where the Input/Output blocks precede line numbering (e.g., Model Immunization from a Condition Number Perspective, ICML 2025).
>
> >Q6: Novelty of each layer.
>
> **R:** SPU is not a standard ensemble but a single, principled architecture tailored to DNA’s multi-scale structure, with LPL/SPL **mechanistically distinct** from ConvNova.
> - *LPL:* ConvNova uses fixed convolution kernels, while LPL generates **token-conditioned mixing weights** to adapt features per input token (Eq. 1).
> - *SPL:* Unlike ConvNova’s generic gating, SPL uses cross-branch, parallel multi-axis gating, providing independent per-dimension control. This dual-axis cross-check filters noise during fusion before it enters the representation, which makes it effective for noisy DNA signals.
> - For periodicity, our contribution is not “replacing Hyena's FFT with wavelets”, but enabling multi-resolution analysis (MRA): CNNs lack frequency resolution, while FFT loses spatial localization (Lemma 3.1). SPU uses wavelets as an MRA pre-conditioner (log-frequency tiling; Theorem 3.2) plus learnable GC, and empirically recovers periodicity missed by other methods (Fig. 5A).
>
> Overall, the novelty is the unified “multi-level plasticity” framework integrating LPL, DPL, and SPL, designed to jointly capture local motifs, long-range dependencies, and structural periodicity without the trade-offs inherent to existing models.
>
>
> >Q7: Implementation details.
>
> **R:** Yes, the token-specific coefficients are passed through a softmax to stabilize feature fusion. The global convs do not share parameters, allowing each frequency band to learn independent patterns.

---

> > ### Author Rebuttal · Reviewer_7GHL · 2026-04-03
> >
> > I appreciate the authors' effort. However, since my major concerns remain unresolved, I will maintain my score.
> >
> > While the authors claim a DNA-tailored framework, the design is overly complex and lacks fundamental novelty. Modules such as branching, gating, and multi-resolution processing are not inspiring in 2026, and stacking them together does not make a big contribution.
> >
> > Besides, it is mainly validated on tasks that are disconnected from what the scientific community actually cares about, making it hard to believe this work will have broader impact.
> >
> > Also, I do not understand why evaluating on the revised NT benchmark would confound the comparison.

---

> > > ### Author Response · Authors · 2026-04-03
> > >
> > > Thanks for the reviewer's feedback. We respectfully, yet fundamentally, disagree with the assessment regarding our novelty, task relevance, and benchmark selection. We address these points directly below:
> > >
> > > 1. SPU is a **theoretically grounded, principled, and biologically-driven** architecture: We clarify that SPU is not a mere stacking of existing modules, but a highly synergistic architecture explicitly tailored to the unique topology of DNA. As demonstrated in our experiments, each layer serves a distinct, biologically motivated purpose that fundamentally differs from existing approaches:
> > >     - LPL: Flexibly captures local functional elements across diverse sequence contexts.
> > >     - DPL: Fuses time- and frequency-domain signals, effectively modeling the long-range dependencies and periodicity intrinsic to DNA.
> > >     - SPL: Uniquely designed to activate functional "hotspots" while aggressively suppressing the massive uninformative background noise (i.e., junk DNA).
> > >
> > > Importantly, these highly complementary modules do not introduce severe computational overhead. As evidenced by our scaling benchmarks in the rebuttal, SPU maintains quasi-linear efficiency and outperforms Transformer, Hyena and ConvNova in ultra-long sequence processing.
> > >
> > > 2. We respectfully disagree that our evaluated tasks are disconnected from the community's interests. The core objective of this paper is to introduce and validate a **foundational computational operator for DNA modeling** (akin to Hyena, Bio-LSTM, Transformer), rather than a **vertical, task-specific application model** (like EVO2, AlphaGenome).
> > > To rigorously measure the fundamental sequence modeling capabilities and intrinsic representational capacity, we evaluated SPU on the exact comprehensive benchmarks (**covering short-range, long-range [up to 131K length in our original paper], and periodic scenarios**) established by leading DNA architectures (**including recent works such as ConvNova 2025, Bio-LSTM 2025**). Advancing the expressivity and efficiency of foundational operators is the critical prerequisite for empowering stronger downstream vertical models, which directly drives the scientific community forward.
> > >
> > > 3. Evaluating on the revised NT benchmark would confound our comparative analysis. The state-of-the-art baselines we compared against (Hyena, Caduceus, ConvNova) provide official, highly tuned hyperparameters for the original NT benchmark specifically. Forcing these baselines onto a new data distribution without official tuning guidelines would compromise reproducibility and fundamentally violate a fair, apples-to-apples comparison.

---

### Official Review · Reviewer_GWWQ · 2026-03-12

**Soundness:** 3
**Presentation:** 3
**Significance:** 4
**Originality:** 3
**Overall Recommendation:** 5
**Confidence:** 3

**Summary:**

The paper introduces the Synergistic Plasticity Unit (SPU), a novel neural network building block for DNA sequence modeling that addresses the core limitations of existing architectures — CNNs, Transformers, and State Space Models — which individually struggle to capture local motifs, long-range dependencies, and periodic biological signals simultaneously. SPU unifies three specialized components within a quasi-linear computational framework: the Locus Plasticity Layer (LPL), which models fine-grained local patterns through token-specific combinations of multi-scale convolutional kernel bases; the Domain Plasticity Layer (DPL), which integrates time- and frequency-domain information via FFT-accelerated global convolution and multi-level wavelet decomposition/reconstruction; and the Saliency Plasticity Layer (SPL), which refines channel and positional importance scores. SPU-based models (SPU-DNA) achieve state-of-the-art or competitive performance across multiple genomic benchmarks — including chromatin profile prediction and variant effect analysis — while scaling efficiently and outperforming larger models. The work is further supported by theoretical grounding in time-frequency localization and interpretability analyses such as power spectral density on nucleosome periodicity and sparse feature disentanglement, establishing a new paradigm for genomic modeling that prioritizes both architectural flexibility and biological interpretability.

**Compliance With Llm Reviewing Policy:**

Affirmed.

**Final Justification:**

The rebuttals answered my questions satisfactorily.

**Key Questions For Authors:**

1. What wavelet family, boundary conditions, and down/up sampling specifics are used in the DPL? How sensitive are results to these choices and do you observe trade-offs in periodicity detection?

2. For the Global Convolution operator, please clarify the exact conditioning mechanism (AdaLN on what statistics? Across channels/positions?) the shapes of z and h, and any regularization to prevent unstable kernels or aliasing during FFT-based convolution.

3. The SPL appears similar to CBAM’s channel+spatial attention. Did you compare against a 1D CBAM baseline with matched capacity? If not, could you add this comparison to isolate the advantage of SPL’s design?

4. Can you report sequence-length scaling experiments (12k, 50k, 100k, 200k) for throughput and accuracy to empirically validate quasilinear behavior and the long-range benefit?

**Limitations:**

While authors include an Impact Statement, they do not adequately discuss the potential negative societal impacts or technical limitations of their work, concluding instead that the work “poses no foreseeable negative risks to society”.

The authors briefly mention that the model trails mature State Space Models in runtime due to the absence of custom CUDA kernels. A more detailed discussion on the engineering efforts to bridge this gap or the stability of the input-dependent Global Conv. kernels over extremely long sequences would be beneficial.

**Strengths And Weaknesses:**

*Soundness: 3*
1. The derivation of the “Static Resolution Limitation” (Lemma 3.1) correctly identifies the trade-offs in CNNs and global models. The logarithmic multiresolution tiling proves that the DPL constructs a MRA structure. This Heisenberg Uncertainty Principal analysis is largely descriptive of established MRA properties, however while it justifies why a wavelet based coverage is superior to static tiling, it does not provide a mathematical guarantee that the specific learnable operators within the SPU will generalize better or optimize more efficiently than standard wavelets or other spectral methods. It acts as a strong motivator for the architecture of the model but lacks of providing a new theorem for learnable DNA operators.

2. The design of the Domain Plasticity Layer (DPL) aligns with best practices of signal processing. The decomposition and reconstruction using Global Convolution-per-band is technically sound, allowing the model to learn distinct patterns within specific frequency resolutions. The recursive Inverse Wavelet Transform fusion is well-motivated by its inclusion of skip connections which helps to preserve original low-freq. Information and prevent signal degregation during reconstruction. Algorithm 1 introduces a data-dependent kernel, but several technical specifics are underspecified:

        a. The paper state AdaLN is “conditioned by $F_{lpl}$” but don’t detail the transformation mechanism.

        b. Beyond a simple cropping step in Step 14, to align the sequence, the paper does not discuss how it mitigates the periodic wrap-around artifacts in FFT based convolutions.

        c. The kernel $p$ is modulated by an exponential decay which may provide some natural regularization, the paper lacks a stability analysis to ensure the data dependent MLP-generated filters don;t lead numerical explosions over long sequences

3. The Saliency Plasticity Layer (SPL) uses dual scoring where Channel Saliency uses average pooling and MLP and Position Saliency uses average/max pooling and convolution. This scoring is nearly identical in spirit to CBAM. (https://arxiv.org/pdf/1807.06521)

4. While the authors specify a 7x7 kernel for depthwise separable convolution in the SPL, they do not provide an ablation for why this specific design or the use of depthwise seperable layers is better for 1D genomics compared to classic attention mechanisms. The claim is that SPL is more comprehensive than locally gated approaches is supported by performance but the specific design choices are not based for their biological relevance.

5. The model’s empirical results provide significant evidence of its utility. SPU-DNA established new SOTA results on 7 out of 8 genomic benchmark datasets and 16 out of 18 Nucleotide Transformer tasks. The clear scaling law suggests the architecture is robust and benefits from increased parameter size. The use of a Sparse Autoencoder to prove that pretrained embeddings align with promoters and CDS regions provides a layer of interpretability that validates the model is learning real biological structures rather than just exploiting dataset biases.

 The submission is technically sound as high-performance engineering contribution where its unique spectral sensitivity is empirically superior to existing models. However paper’s theoretical novelty is descriptive and its implementation details (specifically concerning Global Conv. stability and SPL’s distinction from CBAM) require more transparency.

*Presentation: 3*

1. High level motivation and architectural overview are clear. The decomposition reconstruction path in DPL is illustrated and equations are provided for the reconstruction steps.

2. The theoretical section frames the design via time-freeq localization and Heisenberg-style arguments, which helps connect engineering choices to principled considerations.

3. However, some algorithmic details for GC and DPL are underspecified. For example, precise wavelet family, boundary handling, padding/cropping, and parametrization of decay/shift in GC.

4. Missing key references and comparisons: CBAM ( channel and spatial attention), scattering transform, wavelet CNNs, frequency-aware spectral blocks…

5. Recent critiques on long-context benefits and confounding in genomics such as Prism are not discussed. The paper would benefit from engaging with this perspective.

*Significance: 4*

1. The paper addresses a significant and well-documented challenge in genomics which is the need for an architecture capable of simultaneously capturing local motifs, long range dependencies and periodic biological signals. By introducing the SPU the authors advance machine learning practice through a design that integrates time and frequency domains where a feature lacking in existing CNN, Transformer and State Space Model (SSM) architectures.

2. The paper identifies that current models operate exclusively in the time domain, often failing to represent important periodic signals such as nucleosome positioning or codon periodicity.

3. The SPU achieves O(L log_2 L) complexity which is essential for genome-scale sequences where the quadratic complexity of Transformers is prohibitive. Benchmarks show it achieves the lowest FLOPs compared to Mamba and Hyena for 100k sequence lengths while maintaining superior performance on 16 out of 18 datasets in the Nucleotide Transformer Benchmark.

*Originality: 3*

1. SPU-DNA is a novel genomic foundation model designed to overcome the limitations of standard architectures by integrating local motifs, long-range dependencies and periodic signals. The model achieves SOTA performance on both short and long range genomic benchmarks while operating with O(L log_2 L) complexity, making it highly efficient alternative to Transformers for ultra-long sequences.

---

> ### Author Rebuttal · Authors · 2026-03-31
>
> Thank you for your thorough review and constructive suggestions. We address each point below.
> > Q1: MRA bounds & learnable operators.
>
> **R:** While deriving strict generalization bounds for non-convex deep nets remains an open problem, approximation theory implies SPU has strictly stronger expressivity. Let $H_{wav}$ be the hypothesis space of standard wavelets. Since SPU’s data-dependent Global Conv (GC) can emulate any fixed wavelets, its hypothesis space strictly contains them: $H_{wav} \subseteq H_{SPU}$. This yields two implications:
> 1. Lower bound on error: SPU avoids the irreducible mismatch error of fixed wavelets when biological motifs deviate from preset wavelet shapes;
> 2. Higher approximation capacity: given the optimal MRA time–frequency tiling (Theorem 3.2), the learnable GC provides a higher approximation capacity within each optimal tile for diverse DNA patterns.
>
> We'll add this analysis to make the theoretical contribution explicit.
> > Q2: Alg. 1 details.
>
> **R:** a) For AdaLN, we apply *GAP* on $F^{lpl}$ to obtain a sequence-level descriptor, then use a linear projection to generate $(\gamma,\beta)$ for modulating.
> b) Before the FFT, we zero-pad sufficiently so the resulting circular convolution matches the desired linear convolution on the valid range. We then crop to the valid interval, discarding padded boundaries, ensuring artifact-free outputs.
> c) Numerical stability (BIBO guarantee): Our filter is BIBO-stable by construction. First, MLP outputs are normalized (e.g., LayerNorm), which bounds the raw kernel amplitude $\max|h|\le H_{\max}$ and avoids spurious spikes. Second, the modulation term $\exp(-t\otimes|\delta|)$ imposes strict geometric decay, yielding a length-independent $L_1$ bound:
> $ \sum_{t=0}^{L-1}|p(t)| \le \frac{H_{\max}}{1-\exp(-|\delta|)}$. For $|\delta|>0$, this bound is finite and independent of $L$, preventing numerical and gradient explosions on ultra-long sequences.
> > Q3: SPL vs. CBAM & why use convs with ksize=7?
>
> **R:** SPL is designed for DNA and differs from vision CBAM in both structure and motivation:
> 1. *Parallel gating:* CBAM sequentially refines the *same* feature via channel→spatial attention; SPL computes saliency $(S_c, S_p)$ in parallel (with independent per-dimension control) and uses it to gate DPL time–frequency features, enabling cross-branch fusion.
> 2. *Operator design:* CBAM uses regular convs; SPL uses a *1D depthwise separable conv with kernel size 7* to match typical functional elements (e.g., TFBS for ~6–10 bp). This operator also localizes motifs efficiently on long sequences without the $\mathcal{O}(L^2)$ cost of classic attention mechanisms.
> 3. *Robust scoring:* CBAM uses max pooling; SPL uses *average pooling only*, as max pooling is sensitive to outlier spikes in non-stationary DNA signals.
>
> Empirically, swapping SPL with CBAM on the Genomic Benchmark reduces mean accuracy across 8 datasets by 1.03%, supporting SPL’s better fit for DNA modeling.
>
> >Q4: More technique details.
>
> **R:** SPU uses Haar (db1) with symmetric padding; GC zero-pads to $2L$ to avoid wrap-around and crops back to $L$ after IFFT, learning decay $\delta$ (init 0.1) and shift $s$ (init 0) end-to-end; z has shape $(B,L,D)$ and h has shape $(B,D,L)$. Results are largely *insensitive* to the wavelet settings since WT serves only as an MRA pre-conditioner, while the learnable GC handles periodicity. For periodicity trade-offs, higher-order wavelets isolate frequencies more smoothly but blur short-motif locations; we use Haar to preserve sharp spatial boundaries, and let the data-dependent GC refine periodic signals. The runtime gap to SSMs mainly comes from torch.fft HBM I/O; we are developing FlashFFTConv-style fused kernels to improve efficiency. We'll open-source the code upon acceptance.
> >Q5: Missing references.
>
> **R:** We'll add these references and clarify SPU’s key distinctions. (E.g., unlike Wavelet CNNs, SPU uses wavelets as an MRA pre-conditioner and then applies learnable GC (not fixed wavelet features/downsampling); unlike spectral blocks, SPU combines wavelet tiling with GC to capture long-range periodicity while preserving spatial motif localization; addressing Prism’s long-context noise concern, SPU uses SPL to dynamically gate noise (Fig. 4), enabling robust ultra-long-context modeling.)
> >Q6: Length scaling experiments.
>
> **R:** We ran the requested scaling experiments up to 1M tokens (we kindly ask you to refer to the results in our response to *Reviewer GB8B, Q3*). Key takeaways:
> 1) *Quasi-linear throughput:* SPU exhibits $O(L)$ memory and $O(L\log L)$ compute, and is consistently faster and FLOP-efficient than Hyena and Transformer, demonstrating megabase feasibility competitive with SSMs.
> 2) *Long-range accuracy:* SPU achieves the *best VEP AUROC*. Mamba’s speed advantage comes from *lossy state compression* that hurts long-range accuracy. SPU avoids these trade-offs, yielding stronger long-context modeling.

---

> > ### Author Rebuttal · Reviewer_GWWQ · 2026-04-01
> >
> > I appreciate the sincere clarifications to my questions. I will maintain my original score.

---

> > > ### Author Response · Authors · 2026-04-02
> > >
> > > We sincerely thank you for your **positive support** of our work, and your valuable feedback has further improved the quality of our work.

---

### Official Review · Reviewer_zt3e · 2026-03-13

**Soundness:** 3
**Presentation:** 3
**Significance:** 2
**Originality:** 2
**Overall Recommendation:** 3
**Confidence:** 3

**Summary:**

introduces a new neural network architecture called the Synergistic Plasticity Unit (SPU) designed specifically for analyzing DNA. The authors argue that current AI models like Transformers or CNNs are too rigid because they often focus only on local patterns or only on global sequences. To fix this, the SPU uses three specialized layers: one for identifying tiny local "motifs" (Locus Plasticity), one for looking at both the timing and the frequency of patterns across the whole sequence (Domain Plasticity), and one for filtering out unimportant data (Saliency Plasticity)

**Compliance With Llm Reviewing Policy:**

Affirmed.

**Key Questions For Authors:**

how the model handles DNA sequences that have significant amounts of "junk" or non-coding regions that don't follow clear periodic patterns. It would also be helpful to know if there is a specific biological reason why the three-layer structure was chosen over a more unified design.

**Limitations:**

it has not yet been tested on "single-cell" data, which is much noisier and more complex than the bulk genomic data used in the benchmarks.

**Strengths And Weaknesses:**

Strength: innovative multi-domain approach and its impressive performance across different biological tasks. By looking at DNA in both the time and frequency domains, the model can catch periodic signals that other models completely miss. The researchers tested the SPU on a wide variety of benchmarks, ranging from genomic track datasets to species classification, and it consistently outperformed established models like Mamba and Hyena while using fewer parameters. Another strong point is the model's efficiency; the Saliency Plasticity Layer acts as an "information gatekeeper," ensuring the model doesn't waste energy processing noisy or irrelevant parts of the DNA sequence.


Weak:
First, the term "plasticity" is used very broadly throughout the text, and while it sounds impressive, the paper doesn't strictly define how this mathematical "plasticity" relates to the biological concept of plasticity in a way that is easy for non-experts to follow. Second, the model’s complexity is a concern; because it uses three distinct layers with different operations like convolutions and Fast Fourier Transforms (FFTs), it might be difficult for other researchers to implement or integrate into existing software pipelines. Third, although the model is efficient in terms of parameters, the paper does not provide a deep dive into the actual "wall-clock" training time compared to simpler models, which is a critical factor for labs with limited computing power. Fourth, the evaluation mostly focuses on short-to-medium length sequences, leaving it unclear if the SPU can handle the extremely long, million-base-pair "megabase" scales that some modern genomic research requires.

---

> ### Author Rebuttal · Authors · 2026-03-31
>
> Thank you for your thorough review and constructive suggestions. We address each point below.
> > Q1: Plasticity related to biological concept.
>
> **R:** In Lines 10–13 and 20–26 of the paper, we briefly describe the biological motivation for each plasticity layer. We will add a dedicated paragraph to explicitly clarify how SPU’s mathematical “plasticity” relates to the biological notion of plasticity:
> *Locus Plasticity (short-term adaptation):* The functional motifs are important elements in genomics, so we use token-specific dynamic weights (Eq. 1) to effectively adapt the receptive field to local DNA context.
> *Domain Plasticity (aperiodic–periodic fluidity):* Genomes contain both aperiodic 1D patterns and periodic signals (e.g., codon/nucleosome repeats). Domain Plasticity dynamically routes and fuses features between the time domain (aperiodic) and frequency domain (periodic) to match the underlying biological signal.
> *Saliency Plasticity (hotspot activation):* Functional elements are sparse and context-dependent, embedded within large non-functional regions. Saliency Plasticity applies parallel multi-axis scoring to gate information flow, amplifying salient regulatory hotspots while suppressing background noise.
>
> >Q2: Implementation complexity.
>
> **R:** SPU is built entirely from standard, highly optimized primitives available in modern deep learning frameworks. More importantly, SPU is a plug-and-play backbone module that is more efficient than Transformers (see Lines 426-436 and Appendix B) and remains highly competitive in efficiency compared with other models (see Table 5). Upon acceptance, we will fully open-source the code, enabling SPU-DNA to be integrated with only a few lines of code.
>
> >Q3: Wall-clock for limited compute.
>
> **R:** Table 5 compares SPU with highly efficient methods (e.g., Mamba), showing that SPU is both practical and competitive. We also benchmark a simple fully convolutional model at 100k length for training, ConvNova: it requires 15,327 GFLOPs, 13.8 GB VRAM, and 2.53 s wall-clock time, making it less efficient than both Mamba and SPU. By contrast, SPU processes a 100k-token sequence in 0.93 s per forward/backward pass using only 6.8 GB VRAM, making long-sequence DNA modeling feasible for labs without large compute clusters. We further estimate that pretraining an 0.5B SPU-DNA model on 4 RTX 4090 GPUs for 1 epoch over hg38 with 10k-length sequences would take 5–6 hours, which is relatively fast.
>
> >Q4: Handling "megabase" scale sequences.
>
> **R:** SPU can handle long-range sequences efficiently. In variant effect prediction experiments, it processes sequences up to 131k base pairs and outperforms Enformer and NT-V2. We further benchmarked training efficiency on 1M-token sequences. Mamba uses 57,836 GFLOPs, 61 GB VRAM, and 5.2 s wall-clock time; Hyena uses 75,392 GFLOPs, 72 GB VRAM, and 15.3 s; SPU uses 56,931 GFLOPs, 66 GB VRAM, and 9.9 s. A Transformer cannot run for this length on our GPU. Overall, SPU is markedly more efficient than Hyena and Transformers at 1M tokens, achieves the lowest FLOPs among the tested models, and remains highly competitive.
>
> > Q5: Handling "junk" DNA & the 3-layer biological rationale.
>
> **R:** SPU effectively handles “junk DNA,” and its multi-layer design has a clear biological rationale:
> 1) *Non-coding (“junk”) DNA:* Mechanistically, SPU does not force periodic structure. In DPL, learnable band-wise scaling can down-weight frequency features when periodic cues are absent, while SPL further gates out irrelevant activations. Appendix R (Table 15) shows SPU does not hallucinate spurious periodic signals on non-periodic inputs. Furthermore, GB and NT benchmarks contain many non-coding regulatory sequences, and SPU’s strong results on them indicate it is not derailed by unstructured background.
>
> 2) *Why a 3-layer design:* A unified, homogeneous module struggles with DNA’s heterogeneous, multi-scale regulation. SPU uses an explicit inductive bias: *LPL* captures local motifs, *DPL* models long-range and time–frequency structure, and *SPL* selectively amplifies functional hotspots while suppressing background. By decoupling these mechanisms, SPU prevents the destructive interference of unified models. It ensures local motifs (LPL) aren't washed out by broad periodicities (DPL), while cleanly isolating genuine signals from genomic noise (SPL). This tripartite design yields a mathematically and biologically aligned representation.
>
>
> > Q6: Testing on single-cell data.
>
> **R:** We clarify that single-cell data primarily involves transcriptomic (RNA) profiles, while SPU is explicitly a DNA model. The noise and sparsity of single-cell data result from the limited amount of RNA fragments, rather than sequence information. To ensure an apples-to-apples evaluation of a DNA model, we follow the standard genomic benchmarks used by prior DNA models (e.g., DNABERT-2, HyenaDNA, EVO). Extending SPU to noisy single-cell omics is a valuable direction for future work.

---

> > ### Author Rebuttal · Reviewer_zt3e · 2026-04-03
> >
> > I am still concerned about point 4 and 6, even though author mention it might be future direction I remain skeptical about the methods wider application. I choose to remain score

---

> > > ### Author Response · Authors · 2026-04-05
> > >
> > > *We sincerely thank you for the constructive feedback. We fully agree that the broader applicability of SPU must be proven empirically rather than deferred to future work. To explicitly address the remaining concerns and provide further clarity, we have conducted additional, targeted experimental validations, as detailed below:*
> > >
> > > 1. **Further response to point 4:** While our previous response demonstrated SPU's computational advantages at a 1M-token context, we have conducted an additional Variant Effect Prediction (VEP) experiment using extreme sequence lengths (1M tokens) to explicitly address your concerns regarding predictive performance at this scale.
> > > Specifically, we curated data from the Variant Effect subset of the `genomics-long-range-benchmark` (released by InstaDeepAI). We isolated 300 variant loci located strictly >1M tokens away from their respective Transcription Start Sites (TSS). By constructing 1M-length sequences with these variants as inputs, we are uniquely positioned to evaluate both computational efficiency and predictive accuracy on true megabase-scale contexts. The dataset was partitioned into 240 training and 60 testing samples, and we retained AUROC as the evaluation metric. The comparative results are summarized in the table below.
> > > *Results & Analysis:* Transformers run out of memory at this sequence length on our GPU. Compared to other SOTA long-context models (Mamba and Hyena), **SPU successfully achieves the highest AUROC while maintaining the lowest FLOPs.** While Mamba exhibits lower VRAM usage and wall-clock time, this efficiency comes at the steep cost of lossy state compression, which degrades predictive accuracy. Ultimately, this experiment directly validates SPU’s capability to process megabase-length sequences, reinforcing that SPU’s modest runtime overhead versus SSMs is fully justified by its superior biological utility and its capacity for exact long-range modeling; moreover, because SPU also achieves the lowest FLOPs, it likely retains substantial headroom for further acceleration through implementation and hardware optimization.
> > >
> > > *Performance comparison on **"megabase" scale (1M-token length)** sequences*:
> > >
> > > *(Setting for resource benchmarks: ≈12M parameters, an A800 80GB GPU, and one forward–backward pass.)*
> > >
> > > | Model | VRAM (GB) | FLOPs (G) | Wall-clock (s) | VEP (AUROC %) |
> > > | :--- | :--- | :--- | :--- | :--- |
> > > | Transformer + FlashAttn | OOM | N/A | N/A | N/A|
> > > | Mamba | 61.3 | 57,836 | 5.21 | 48.3|
> > > | Hyena | 72.2 | 75,392 | 15.32 | 45.2|
> > > | SPU | 66.1 | 56,931 | 9.93 |51.6|
> > >
> > > 2. **Further response to point 6:** To explicitly demonstrate SPU’s versatility on noisy, complex single-cell data, we designed a single-cell ATAC-seq peak prediction task using the GM12878 cell line. Specifically, we utilized a human (GM12878) and mouse (A20) mixture dataset downloaded from 10x Genomics, directly using the standard filtered matrix. To reliably identify human cells, we computed for each cell the proportion of counts mapping to human peaks, retained cells with a proportion exceeding 90%, and then randomly sampled 100 of the retained human cells for downstream experiments. For positive samples, we extracted sequences at peak locations with non-zero counts for each cell. For negative samples, we randomly sampled sequences outside the consensus peaks.
> > > Following the standard multi-task formulation for single-cell sequence modeling (e.g., scBasset), the models take a 10K-bp DNA sequence as input and output an 100-dimensional vector, simultaneously predicting the peak accessibility state across all the sampled individual cells. To ensure strict spatial isolation, models are trained on chr1-19 and tested on the held-out chromosomes. We report the Average AUROC and AUPRC to account for the extreme sparsity and class imbalance inherent in scATAC data (shown below).
> > > *Results & Analysis:* The severe sparsity and "dropout" artifacts in single-cell data pose extreme challenges for standard sequence models. While the other models struggle to extract robust patterns from such noisy labels, **SPU achieves the highest AUROC (83.8%) and AUPRC (42.5%)**. This confirms that **SPU is not restricted to bulk genomic data**. In fact, SPU's Saliency Plasticity Layer (SPL) acts as a mathematically rigorous noise-suppression mechanism, allowing the architecture to reliably locate functional structural motifs even when the supervisory single-cell signal is highly corrupted by technical dropout.
> > >
> > > *Performance comparison on **single-cell** ATAC peak prediction* (model size: 12M):
> > > | Model | AUROC (%) | AUPRC (%) |
> > > | :--- | :--- |:--- |
> > > | Transformer + FlashAttn | 76.5 | 34.6 |
> > > | Mamba | 81.9 | 40.6 |
> > > | Hyena | 79.6 | 38.8 |
> > > | SPU | 83.8 | 42.5 |
> > >
> > > **We hope that this empirical effort to transform a 'future direction' into an 'immediate reality' directly addresses your core concerns, and we respectfully hope you might reconsider the evaluation of our work in light of these new breakthroughs.**

---

### Official Review · Reviewer_GB8B · 2026-03-13

**Soundness:** 3
**Presentation:** 3
**Significance:** 2
**Originality:** 3
**Overall Recommendation:** 4
**Confidence:** 3

**Summary:**

This paper introduces the Synergistic Plasticity Unit (SPU), a novel building block designed for DNA sequence modeling. The architecture seeks to address the limitations of existing models (CNNs, Transformers, and SSMs) in capturing the multi-scale nature of genomic data, specifically local motifs, long-range dependencies, and structural periodic signals. The SPU integrates three synergistic layers: the Locus Plasticity Layer (LPL) for dynamic local feature extraction, the Domain Plasticity Layer (DPL) for joint time-frequency modeling via wavelets and global convolutions, and the Saliency Plasticity Layer (SPL) for feature refinement. Supported by theoretical analysis based on the Heisenberg Uncertainty Principle, the authors demonstrate state-of-the-art performance across multiple genomic benchmarks while maintaining quasi-linear computational complexity.

**Compliance With Llm Reviewing Policy:**

Affirmed.

**Final Justification:**

The empirical contribution of this paper is solid. SPU-DNA achieves strong results across multiple genomic benchmarks, and the ablation study confirms that each layer contributes meaningfully. The hardware scaling experiments added in the rebuttal (12k to 1M tokens) resolved my efficiency concerns — SPU is practical at megabase scales with competitive FLOPs.

My reservation centers on the theoretical framing. Lemma 3.1 and Theorem 3.2 are technically correct, but the analytical framework motivates multi-resolution processing in general, not SPU's specific three-layer combination in particular. The rebuttal's characterization of this combination as "the mathematically necessary approach" overstates what the proofs establish. I would like the revised paper to distinguish more carefully between the theoretical motivation (general) and the architectural instantiation (domain-specific, one of several valid paths). On the presentation, citing the relevant signal processing references would be more precise and reduce unnecessary cognitive overhead. Grounding the narrative in the mathematical tradition it actually draws from would better serve the paper's clarity.

I also share Reviewer 7GHL's concern that the benchmarks used are outdated — demonstrating results on more current benchmarks would strengthen the paper's relevance.

Overall, I raised my score from 3 to 4 in recognition of the experimental strengths. The paper sits at the borderline: the experiments are above average, but the theoretical narrative needs more careful calibration. I defer to the AC on the benchmark concerns and would need to see the theoretical framing toned down in revision to fully support acceptance.

**Key Questions For Authors:**

- Given the high complexity of the SPU, did the authors experiment with a simplified version of the DPL? Specifically, could a standard Global Convolution (without the 3-level Wavelet Transform hierarchy) achieve similar results if the parameter count were kept equal?
- Does the SPU architecture scale as efficiently as Transformers or SSMs in terms of compute-to-performance ratio? Specifically, as the sequence length increases to $10^6$ or $10^7 $ tokens, how does the memory overhead of the multiple parallel paths in the SPU compare to the state-compression approach of Mamba?

**Limitations:**

The SPU architecture's reliance on multiple parallel paths and hierarchical wavelet decompositions introduces significant engineering complexity and memory overhead, which likely limits its scalability compared to more unified architectures like Mamba or Transformers. Furthermore, the design is heavily tailored to the physical periodicity of DNA, raising concerns about its generality as a foundation model for broader sequence modeling tasks.

**Strengths And Weaknesses:**

### Strengths
- The model effectively incorporates biological priors, such as nucleosome periodicity and regulatory motifs, into the architectural design.
- The SPU-DNA model outperforms several established foundation models, such as HyenaDNA and Nucleotide Transformer, across diverse tasks.


### Weaknesses
- The SPU module is essentially a heuristic ensemble of distinct components, convolutions, wavelet transforms, and saliency mechanisms, linearly stacked together. While this design proves effective for DNA, it lacks the architectural elegance and simplicity expected of a fundamental machine learning unit. Furthermore, the design is heavily biased toward the physical properties of DNA (e.g., periodicity), which limits its transferability to general sequence modeling tasks.
- The theoretical derivation involving the Heisenberg Uncertainty Principle (HUP) and Multi-Resolution Analysis (MRA) primarily restates classical signal processing concepts. While the authors map these to the SPU layers, they fail to prove that this specific complex configuration is the necessary or optimal path for improving DNA modeling. The theory feels like a post-hoc justification for a complex modular assembly rather than a foundational insight that dictated the architecture's design.
- Although the model claims quasi-linear complexity of $O(L \log L)$, the actual hardware efficiency is a concern. The SPU involves numerous parallel branches and complex transformations (FFT, WT), which are often memory-bandwidth intensive and unfriendly to modern GPU scheduling. The authors acknowledge that, without custom CUDA kernels, the actual wall-clock time lags behind mature SSM architectures like Mamba. This gap between theoretical complexity and practical throughput is a significant drawback for a model intended for ultra-long sequence pre-training.

---

> ### Author Rebuttal · Authors · 2026-03-31
>
> Thank you for your thorough review and constructive suggestions. We address each point below.
> > Q1: Heuristic ensemble vs. fundamental design & transferability.
>
> **R:** We respectfully clarify that SPU is designed for DNA modeling, and it is not a heuristic ensemble but a principled architecture derived from the requirements of Multi-Resolution Analysis (MRA). DNA sequences are inherently complex, featuring sharp local motifs, long-range dependencies, and structural periodic signals concurrently. No single operation (like a CNN or transformer) can optimally capture this diversity without sacrificing resolution in either the time or frequency domain. The combination of LPL, DPL, and SPL is the mathematically necessary approach to achieve logarithmic multi-resolution tiling (Section 3.2), allowing the model to adaptively cover the entire signal space. **This core principle is theoretically transferable to other non-stationary sequence data (e.g., audio, ECG) beyond genomics.**
>
> > Q2: Theoretical derivation as post-hoc justification.
>
> **R:** We clarify that theorem 3.2 is not a restatement of classical MRA; it formally shows how the SPU instantiation overcomes the static time–frequency resolution limits of standard architectures. Lemma 3.1 proves that existing designs (CNNs or pure FFT/global mixers) are constrained by the HUP: they either behave as broadband filters that miss global periodicity or as global mixers that blur local motifs—**both critical in DNA**. SPU addresses this by placing learnable global convs inside the multi-scale Heisenberg boxes induced by wavelet decomposition (Eq. 3–4), achieving balanced time–frequency coverage. Table 4 empirically supports this: removing any component disrupts this coverage and consistently degrades performance.
>
> > Q3: Hardware efficiency and wall-clock time, scaling and memory overhead vs. SSMs.
>
> **R:** We agree that quasi-linear theory must translate into real speedups. To assess multi-branch overhead, we benchmark hardware efficiency up to 1M tokens against Flash-Transformers, Mamba (SSM), and Hyena. (Shown below)
>
> 1) *Megabase-scale hardware feasibility:* At 1M tokens, Transformers are blocked by the $O(L^2)$ computation bottleneck and run out of memory, whereas SPU preserves an $O(L)$ memory footprint, requiring only 66.0 GB VRAM with 9.93 s runtime. Moreover, SPU achieves the lowest theoretical FLOPs at this scale among competitive exact long-context models (56,931G vs. 57,836G for Mamba) and outperforms Hyena in both compute and wall-clock efficiency. These results demonstrate that SPU is highly feasible for megabase-level deployment.
>
> 2) *Compute–performance trade-off:* Across all sequence lengths, SPU offers a better compute–accuracy frontier than Transformers and Hyena. Mamba is slightly faster and uses less VRAM via lossy state compression, but this degrades accuracy on complex long-range biological tasks. SPU consistently achieves higher Variant Effect Prediction (VEP) AUROC (distance to TSS > 100k) under a very low FLOP budget, so its modest runtime overhead versus SSMs is justified by better biological utility and exact long-range modeling.
>
> *Compute-to-Performance Comparison*
>
> *(Setting for resource benchmarks: ≈12M parameters, an A800 80GB GPU, and one forward–backward pass.)*
>
> | Seq Length | Model | VRAM (GB) | FLOPs (G) | Wall-clock (s) | VEP (AUROC %) |
> | :--- | :--- | :--- | :--- | :--- | :--- |
> | 12k | Transformer + FlashAttn | 1.5 | 7,431 | 0.08 | 53.3|
> | | Mamba | 1.2 | 673 | 0.06 | 54.6|
> | | Hyena | 1.2 | 615 | 0.07 | 53.8|
> | | SPU (ours) | 1.3 | 522 | 0.07 | 56.1|
> | 50k | Transformer + FlashAttn | 4.2 | 129,267 | 1.22 | 53.8|
> | | Mamba | 3.2 | 2,812 | 0.23 | 55.6|
> | | Hyena | 3.4 | 2,956 | 0.44 | 55.2|
> | | SPU (ours) | 3.4 | 2,558 | 0.45 | 57.3|
> | 100k | Transformer + FlashAttn | 7.8 | 516,931 | 4.87 | 53.1 |
> | | Mamba | 5.8 | 5,619 | 0.46 | 58.5|
> | | Hyena | 6.2 | 5,573 | 0.89 | 57.8|
> | | SPU (ours)| 6.8 | 5,431 | 0.93 | 59.8|
> | 200k | Transformer + FlashAttn | 20.4 | 3,548,371 | 19.50 | 52.7 |
> | | Mamba | 12.1 | 14,722 | 0.98 | 57.9|
> | | Hyena | 14.4 | 15,321 | 2.35 | 57.1|
> | | SPU (ours)| 14.8 | 14,522 | 1.98 | 60.0|
> | 1M *(Stress Test)* | Transformer + FlashAttn | OOM | N/A | N/A | Lack of 1M benchmark|
> | | Mamba | 61.3 | 57,836 | 5.21 | Lack of 1M benchmark|
> | | Hyena | 72.2 | 75,392 | 15.32 | Lack of 1M benchmark|
> | | SPU (ours)| 66.1 | 56,931 | 9.93 | Lack of 1M benchmark|
>
> > Q4: Simplified DPL.
>
> **R:** We evaluated this in Appendix M (Table 10) via an ablation that replaces the full DPL with a long-convolution baseline. Because the wavelet decomposition adds few learnable parameters, this substitution keeps the parameter count essentially comparable (≈1.9M). DPL consistently outperforms long convolution across all settings, showing that the multi-level wavelet hierarchy is necessary. Moreover, the wavelet path introduces minimal overhead and sublinear complexity, so SPU remains computationally efficient (see our response to Q3).

---

> > ### Author Rebuttal · Reviewer_GB8B · 2026-04-04
> >
> > I thank the authors for the detailed responses. The hardware scaling experiments (Q3/Q4) are well-executed — the 12k-to-1M benchmarks and the compute-to-performance comparison are convincing additions that substantiate SPU's practical efficiency. These concerns are resolved.
> >
> > Regarding Q1 and Q2, I want to clarify my remaining reservations. The empirical results and ablations are not in question — Table 4 and Fig. 5(A) provide reasonable evidence that the three plasticity layers are complementary. My concern is about how the theoretical analysis frames the architectural motivation.
> >
> > The paper builds its justification on the time-frequency tradeoff, formalized through the Heisenberg Uncertainty Principle (Section 3.2). Lemma 3.1 shows that a fixed-size CNN kernel or a global FFT mixer each occupies a constrained region of the time-frequency plane. Theorem 3.2 then shows that SPU's wavelet path achieves logarithmic multi-resolution tiling. Both results are technically correct. However, the same analysis logic — that multi-scale, multi-domain processing is needed to cover the time-frequency plane — would equally motivate a range of composite architectures. The theoretical framework explains why multi-resolution processing helps, but the specific choices of LPL, DPL, and SPL are better understood as well-motivated design decisions informed by DNA priors, rather than conclusions that follow inevitably from the theory. I would find the narrative more convincing if the paper were more explicit about this distinction.
> >
> > A minor presentation suggestion: the term "Heisenberg Uncertainty Principle (HUP) (Heisenberg, 1927)" naturally directs readers toward quantum mechanics, yet the time-frequency localization tradeoff used in Section 3.2 is a mathematical result whose rigorous formulation belongs to Fourier analysis and signal processing. Citing the relevant mathematical references (e.g., Gabor 1946, or standard wavelet theory texts) would be more accurate to the actual result being invoked and reduce cognitive overhead for the ML audience. More generally, grounding the theoretical narrative in the precise mathematical tradition it draws from, rather than invoking a more prestigious-sounding origin, would better serve the clarity and accessibility of the paper.
> >
> > Overall I raise my score to 4.

---

> > > ### Author Response · Authors · 2026-04-05
> > >
> > > *We sincerely thank you for raising the score and for providing such insightful feedback. We agree with both of your points and will happily incorporate them into the final manuscript.*
> > > 1. *On the Framing of Theoretical vs. Architectural Motivation:* Your characterization is spot on, and we appreciate this insightful distinction. We fully agree that our mathematical framework (Theorem 3.2) proves the necessity of logarithmic multi-resolution tiling, but does not mathematically necessitate the specific implementations of LPL, DPL, and SPL. To be precise, our model architecture is **jointly driven by MRA theory and biological characteristics**. While a broad range of composite architectures could theoretically satisfy the mathematical requirements of MRA, SPU is specifically designed to satisfy MRA while simultaneously addressing the unique biological priors of DNA. In the revised manuscript, we will explicitly clarify this joint motivation. We will structure the narrative to state that the MRA theory defines the optimal structural constraints (the necessary "search space" of time-frequency tiling), while the distinct biological characteristics of DNA (e.g., varying motif scales, long-range periodicity, and severe non-coding noise) strictly guided the specific instantiation of the LPL, DPL, and SPL from within that space.
> > > 2. *On the Time-Frequency Tradeoff Citations:* We greatly appreciate this presentation suggestion. You are correct that framing the time-frequency localization tradeoff via Heisenberg (1927) unnecessarily drifts into quantum mechanics and creates cognitive overhead for the machine learning and signal processing audience. We will remove the Heisenberg (1927) citation. Instead, we will ground the mathematical tradition strictly in Fourier and signal processing literature by citing 'Gabor 1946' and standard wavelet theory texts. We agree that this adjustment significantly improves the precision and accessibility of the theoretical background.
> > >
> > > Regarding benchmarks, we evaluated our model strictly following the rigorous benchmarks used by SOTA DNA architectures (**including recent works such as ConvNova 2025, Bio-LSTM 2025**). These comprehensively **cover short-range, long-range (up to 131K length in our original paper), periodic, and noisy signals**, ensuring a strictly fair comparison of the fundamental representational capacity of **basic operators for DNA modeling**.
> > >
> > > We hope our further responses address your remaining concerns. We deeply appreciate your rigorous review, which has undoubtedly made our paper stronger and more precise.

---

### Decision · Program_Chairs · 2026-04-30

**Decision:**

Accept (regular)

**Comment:**

The authors propose a new DNA-focused architecture for genomics tasks. The architecture is inspired by the needs of concrete tasks, yet some reviewers found its construction and the underlying theory quite complex. I also found it unclear whether all blocks are necessary and where the major advantages come from. Yet, I do praise the novelty of this paper: clearly, the authors took the time to develop a strategy that targets the genome. I like this; we need more of this. In contrast to reviewers, I am also not concerned about transfer to single-cell or efficiency: clearly, what shines here is performance.
I, however, respect the borderline scores in the paper, yet I suggest acceptance because one reviewer is fully convinced, and myself I also see great value in this.